# Dynamic Regret with Untrusted Decision Predictions via Heterogeneous Expert Aggregation

**Wentao Zhang**
*Tsinghua Shenzhen International Graduate School*
*Tsinghua University*

## Abstract

We study online convex optimization with dynamic regret, where the learner has access to untrusted decision predictions about the per-round minimizers. Existing methods either exploit only gradient feedback, achieving $O(\sqrt{T(1 + P_T)})$ dynamic regret but remaining unable to benefit from predictions, or follow predictions blindly, obtaining regret proportional to the prediction error but with no worst-case safeguard. We propose a framework based on heterogeneous expert aggregation that simultaneously adapts to both the environment non-stationarity, characterized by path length $P_T$, and prediction quality, measured by cumulative error $\tilde{E}_T$, without prior knowledge of either. The framework maintains a diverse pool of experts, which includes a gradient-based expert utilizing Online Gradient Descent, a prediction-based expert following predictions, and a new hybrid subroutine called Online Anchor Mirror Descent. These experts are aggregated by AdaHedge, whose small-loss property is critical to our results. We prove that our strongest variant achieves dynamic regret that smoothly interpolates between $O(GD \log \log T)$ when predictions are accurate and $O(R^*)$ when predictions are adversarial, where $R^* = O(G\sqrt{T(D^2 + 2DP_T)})$ is the optimal prediction-free rate. The small-loss bound of AdaHedge ensures that the aggregation overhead depends on the best expert's loss rather than on $T$, enabling a qualitative improvement over the $\Omega(\sqrt{T})$ floor of prediction-free methods. We further introduce an instance-dependent refinement of the new hybrid subroutine that can strictly improve the guarantee on favorable trajectories. Experiments on synthetic benchmarks validate all theoretical predictions: our methods achieve near-constant regret under accurate predictions, degrade gracefully under adversarial predictions, and outperform baselines by up to $26\times$ in non-stationary environments.

## 1 Introduction

Online convex optimization (OCO) (Hazan, 2016) studies sequential decision-making against adversarial loss functions. While the classical measure of performance, static regret against a single best fixed action, is well understood (Zinkevich, 2003; Hazan et al., 2007), many applications involve non-stationary environments where the optimal action changes over time. In such settings, dynamic regret, which competes against the sequence of per-round minimizers, provides a more appropriate benchmark. Dynamic regret is parameterized by the path length $P_T$, the total movement of the comparator sequence, and has been adopted in online portfolio selection, adaptive control, and real-time resource allocation.

A long line of work has pursued tighter dynamic regret guarantees Zinkevich (2003) first showed that Online Gradient Descent (OGD) achieves $O(\sqrt{T(1 + P_T)})$ dynamic regret with a properly tuned step size, a bound later made adaptive to the unknown $P_T$ by Ader (Zhang et al., 2018) and further improved by Sword (Zhao et al., 2020). Faster rates under strong convexity were established by (Baby & Wang, 2021). All these methods, however, rely solely on gradient feedback; they cannot exploit auxiliary information even when it is available, and their dynamic regret never falls below $\Omega(\sqrt{T})$, even in stationary environments. Separately, the learning-augmented algorithms paradigm (Lykouris & Vassilvitskii, 2021; Purohit et al., 2018) has shown that machine-learned predictions can substantially improve online algorithms when the

Table 1: Comparison of dynamic regret bounds. $R^* = O\big(G\sqrt{T(D^2 + 2DP_T)}\big)$: optimal OGD bound. $\bar{E}_T = \sum_{t=1}^{T} \|\tilde{x}_t - x_t^*\|$: cumulative prediction error.

| Method | Adaptive to $P_T$ | Decision predictions | Guarantee |
|---|---|---|---|
| OGD (Zinkevich, 2003) | No | No | $O(R^*)$; requires known $P_T$ |
| Ader (Zhang et al., 2018) | Yes | No | $O(R^* \log T)$ |
| Sword (Zhao et al., 2020) | Yes | No | $O(R^*)$; convex & smooth |
| FTP (Rutten et al., 2023) | – | Yes | $O(G\bar{E}_T)$; no worst-case guarantee |
| Sword++ (Zhao et al., 2024) | Yes | No | $O(R^*)$; general convex |
| Ours (Hedge) | Yes | Yes | $\min\{O(R^*), O(G\bar{E}_T)\} + O(\sqrt{T}\log\log T)$ |
| Ours (AdaHedge) | Yes | Yes | Between $O(GD\log\log T)$ and $O(R^*)$ |

predictions are accurate, while bounding the degradation when they are not. In continuous optimization, optimistic mirror descent (Rakhlin & Sridharan, 2013) uses gradient predictions to sharpen static regret. Yet these optimistic methods target static regret with gradient-level hints, not dynamic regret with decision-level predictions. A natural alternative is to simply follow the decision predictions, referred to as the Follow-The-Prediction (FTP) strategy (Antoniadis et al., 2023), which yields dynamic regret proportional to the prediction error but provides no worst-case safeguard when predictions are poor. Existing dynamic regret methods do not appear to simultaneously exploit decision predictions when they are informative and maintain robust guarantees when they are unreliable.

In this paper, we address this gap through heterogeneous expert aggregation. The idea is intuitive: rather than committing to a single strategy, we maintain a diverse pool of experts, including some purely gradient-based instances (OGD with various step sizes), one purely prediction-based strategy (FTP), and several that blend both signals via a new subroutine called Online Anchor Mirror Descent (OA-MD). A meta-algorithm then adaptively shifts weight toward whichever expert performs best. Our strongest results arise from using AdaHedge (De Rooij et al., 2014) as the meta-algorithm, whose small-loss property ensures that the aggregation cost depends on the best expert's loss rather than on the time horizon $T$. When predictions are accurate, the prediction-based expert dominates and the overhead becomes negligible; when predictions fail, the gradient-based experts take over. Table 1 summarizes the comparison with prior methods.

Existing approaches each capture only one side of the problem. Prediction-free methods are robust, but they inevitably hit a lower bound of $\sqrt{T}$ and thus fail to capitalize on favorable predictive structure. In contrast, FTP can exploit accurate predictions aggressively, but offers no protection when those predictions are misspecified. Our AdaHedge-based approach closes this gap by providing a smooth interpolation between the prediction-optimal and worst-case regimes. It achieves $O(GD\log\log T)$ dynamic regret under accurate predictions while degrading gracefully to within a small constant factor of $R^*$ under adversarial predictions. This ability to move continuously between aggressive exploitation and robust fallback is the main technical advantage of our framework. Our main contributions are as follows:

- We propose a heterogeneous expert aggregation framework for online convex optimization with untrusted decision predictions, which adapts simultaneously to unknown environment non-stationarity and prediction quality without requiring prior knowledge of either.

- We prove that using AdaHedge yields a smooth interpolation between the prediction-optimal and worst-case regimes: the dynamic regret becomes $O(GD\log\log T)$ under accurate predictions and degrades gracefully to a near-optimal prediction-free bound $O(R^*)$ when predictions are poor. The key technical ingredient is the small-loss property, which makes the aggregation overhead depend on the best expert's loss rather than on $T$.

- We introduce Online Anchor Mirror Descent (OA-MD), a hybrid update that treats predictions as soft anchors and provides an additional instance-dependent refinement, and we empirically validate all major theoretical predictions on synthetic non-stationary benchmarks.

## 2   Related Work

### 2.1   Dynamic Regret in Online Convex Optimization

Dynamic regret was first studied by Zinkevich (2003), who showed that OGD attains $O(\sqrt{T}(1 + P_T))$ for general convex losses (which reduces to $O(\sqrt{T(1 + P_T)})$ with optimally tuned step size when $P_T$ is known). Hall & Willett (2013) and Jadbabaie et al. (2015) subsequently tightened this to $O(\sqrt{T(1 + P_T)})$ through optimized step sizes, though their bounds require prior knowledge of $P_T$. To achieve adaptivity, Zhang et al. (2018) proposed Ader, a meta-learning scheme aggregating OGD copies via Hedge, attaining $O(\sqrt{T(1 + P_T)} \log T)$ parameter-free. Sword (Zhao et al., 2020) achieved the optimal rate for convex and smooth losses, and Sword++ (Zhao et al., 2024) extended this to general convex losses. Under stronger curvature, Mokhtari et al. (2016) achieved $O(P_T)$ for strongly convex and smooth losses, and Baby & Wang (2021) derived minimax-optimal rates under various comparator regularity conditions. Another direction pursues strongly adaptive bounds that hold uniformly over all sub-intervals (Daniely et al., 2015). Despite these advances, these methods are prediction-free: they exploit only gradient feedback and cannot use side information about the comparator sequence. As a result, they do not yield sub-$\sqrt{T}$ rates even in stationary environments. In contrast, our framework incorporates decision predictions as first-class inputs and achieves $O(GD \log \log T)$ when predictions are accurate, while maintaining worst-case guarantees within a constant factor of $R^*$.

### 2.2   Online Learning with Predictions and Advice

The algorithms with predictions paradigm formalizes the consistency-robustness trade-off for online algorithms augmented with machine-learned advice. Lykouris & Vassilvitskii (2021) introduced this framework for online caching, and Purohit et al. (2018)extended it to scheduling and ski rental. The paradigm has since been applied to metrical task systems (Antoniadis et al., 2023) and numerous combinatorial problems; see Mitzenmacher & Vassilvitskii (2022) for a survey. In continuous optimization, Rakhlin & Sridharan (2013) laid the foundation for optimistic online learning, where gradient predictions reduce static regret from $O(\sqrt{T})$ to a prediction-error-dependent quantity. Chiang et al. (2012) connected this to proximal-point methods, and Mokhtari et al. (2016) studied optimistic algorithms under varying prediction quality notions. However, these optimistic approaches operate in the static regret setting with gradient-level hints, which is a different regime from ours. More recently, Bhaskara et al. (2020) and Campolongo & Orabona (2021) have begun incorporating side information into dynamic regret analysis, but these works either impose structural assumptions on the predictions or provide guarantees that do not smoothly interpolate between consistency and robustness. Our method differs in that it treats predictions as entirely untrusted and achieves a smooth trade-off through expert aggregation rather than algorithmic modification.

Learning-augmented frameworks have also been extended to decentralized and networked settings. Li et al. (2024) studied learning-augmented decentralized online convex optimization in networks, where agents communicate over a graph and incorporate predictions to improve both static and dynamic regret. While their work also combines predictions with gradient-based updates, it addresses a different setting: decentralized multi-agent optimization with communication constraints, where the goal is to achieve network consensus while using local predictions. Our work considers the single-agent centralized setting and focuses on achieving a smooth consistency-robustness interpolation for dynamic regret through heterogeneous expert aggregation with small-loss bounds.

### 2.3   Expert Aggregation and Small-Loss Bounds

Our meta-algorithm builds on the classical prediction-with-expert-advice framework (Cesa-Bianchi & Lugosi, 2006). The Hedge algorithm achieves $O(\sqrt{T \ln K})$ meta-regret over $K$ experts, which is worst-case optimal

(Cesa-Bianchi & Lugosi, 2006). A central refinement for our purposes is the small-loss bound, which reduces the meta-regret to $O(\sqrt{L^{(k^*)} \ln K})$ when the best expert's cumulative loss $L^{(k^*)}$ is small. This was made fully adaptive by AdaHedge (De Rooij et al., 2014) through a data-driven learning rate. Small-loss bounds have found applications in bandits and tracking the best expert . In the dynamic regret context, Zhang et al. (2018) employed Hedge-style aggregation but used only the standard $O(\sqrt{T \ln K})$ bound, resulting in a $\sqrt{T}$-dependent overhead. Our work demonstrates that upgrading to AdaHedge's small-loss bound enables a qualitative leap: the aggregation cost shrinks with the best expert's loss, which is precisely what allows accurate predictions to drive the overall dynamic regret down to $O(\log \log T)$. The broader insight is that combining heterogeneous experts, such as mixing prediction-based and gradient-based strategies, with a small-loss aggregator naturally yields adaptive consistency-robustness guarantees. To our knowledge, this combination has not been explored in the dynamic regret literature.

## 3  Problem Setup

We consider online convex optimization over a compact convex decision set $\mathcal{K} \subset \mathbb{R}^d$ with diameter $D = \sup_{x,y \in \mathcal{K}} \|x - y\|$. The game proceeds over $T$ rounds between a learner and an adversary. In the interaction protocol, at each round $t \in [T]$, the learner first receives side information $\tilde{x}_t \in \mathcal{K}$ and then selects a decision $x_t \in \mathcal{K}$. The adversary subsequently reveals a convex loss function $f_t : \mathcal{K} \to \mathbb{R}$, and the learner observes a subgradient $g_t \in \partial f_t(x_t)$. The side information $\tilde{x}_t$ is available before the learner commits to $x_t$, but we make no assumptions on its quality. It may be generated by a machine learning model, a domain expert, or any external source, and may be accurate in some rounds while highly misleading in others.

### 3.1  Assumptions

**Assumption 1.** *The set $\mathcal{K} \subset \mathbb{R}^d$ is nonempty, compact, and convex, with $D < \infty$.*

**Assumption 2.** *For every $t \in [T]$, the function $f_t : \mathcal{K} \to \mathbb{R}$ is convex.*

**Assumption 3.** *There exists $G > 0$ such that $\|g\| \le G$ for all $g \in \partial f_t(x)$, $x \in \mathcal{K}$, and $t \in [T]$.*

We only require the side information to be feasible, i.e., $\tilde{x}_t \in \mathcal{K}$ for all $t \in [T]$. Assumptions 1, 2, and 3 are standard in OCO (Hazan, 2016; Shalev-Shwartz, 2025) and imply that each $f_t$ is $G$-Lipschitz on $\mathcal{K}$. Since $f_t$ is convex and continuous on the compact set $\mathcal{K}$, a minimizer exists; we let $x_t^* \in \arg\min_{x \in \mathcal{K}} f_t(x)$ denote an arbitrary one. The feasibility requirement on $\tilde{x}_t$ places no restriction on $\|\tilde{x}_t - x_t^*\|$, which may be as large as $D$ in any given round.

### 3.2  Performance Measure

We evaluate the learner by its dynamic regret, defined as

$$\mathrm{DReg}_T = \sum_{t=1}^{T} [f_t(x_t) - f_t(x_t^*)].$$

Two quantities govern the difficulty of the problem. The path length $P_T = \sum_{t=1}^{T-1} \|x_t^* - x_{t+1}^*\|$ measures the non-stationarity of the environment, where a larger $P_T$ corresponds to a more rapidly changing comparator sequence. The cumulative prediction error $\bar{E}_T = \sum_{t=1}^{T} \|\tilde{x}_t - x_t^*\|$ captures the cumulative discrepancy between the side information and the per-round minimizers. A value of $\bar{E}_T = 0$ means the predictions are perfect, while a large $\bar{E}_T$ indicates that the predictions may be highly inaccurate overall. Neither $P_T$ nor $\bar{E}_T$ is known to the learner in advance. For convenience, we define $R^* := G\sqrt{T(D^2 + 2DP_T)}$, which is the dynamic regret attained by OGD with a properly tuned step size (Zinkevich, 2003) and serves as the standard prediction-free benchmark.

We seek algorithms whose dynamic regret adapts simultaneously to both $P_T$ and $\bar{E}_T$ without prior knowledge of either. Concretely, the algorithm should exploit small prediction error to obtain regret bounds sharper than $R^*$ (consistency), while ensuring that the regret remains comparable to $R^*$ when $\bar{E}_T$ is large (robustness).

## 4    Algorithms

Our approach is built on a simple observation: no single algorithm can simultaneously guarantee near-optimal worst-case dynamic regret and exploit accurate predictions, because the quality of predictions is unknown in advance. We note that this is an intuitive motivation rather than a formal impossibility result; any algorithm that commits to a fixed trade-off between gradient feedback and predictions cannot adapt to unknown prediction quality, and our framework sidesteps this by maintaining multiple strategies and selecting among them adaptively. We therefore construct a pool of heterogeneous experts, each suited to a different regime, and aggregate their decisions with an online meta-algorithm that automatically concentrates weight on the best-performing expert. This section describes the three types of base experts (Section 4.1), the meta-algorithm (Section 4.2), and the overall framework (Section 4.3).

### 4.1    Base Experts

**Online Gradient Descent (OGD).** The first type of expert ignores the side information entirely and performs a standard projected gradient step:

$$x_{t+1} = \Pi_{\mathcal{K}}(x_t - \eta g_t),$$

where $\eta > 0$ is the step size. OGD with a properly tuned $\eta$ attains dynamic regret $R^*$, but the optimal step size depends on the unknown path length $P_T$. To cover the range of possible $P_T$ values, we instantiate $K_A = \lceil \log_2 T \rceil + 1$ copies of OGD with geometrically spaced step sizes $\eta_i = 2^{-i}D/G$ for $i = 0, 1, \ldots, K_A - 1$. We refer to these as Type-A experts. This discretization ensures that the grid contains a step size at the correct scale of the unknown optimum.

**Follow-The-Prediction (FTP).** The second type of expert ignores gradient feedback and directly follows the side information: $x_t = \tilde{x}_t$. FTP achieves dynamic regret $G\bar{E}_T$ by the Lipschitz property of $f_t$, which is small when predictions are accurate but can be vacuously large otherwise. We include a single FTP instance as the Type-B expert.

**Online Anchor Mirror Descent (OA-MD).** The two expert types discussed above represent opposite extremes, where OGD uses only gradients and FTP uses only predictions. To address whether one can blend both signals within a single update, we introduce Online Anchor Mirror Descent (OA-MD). One option is to directly mix the OGD and FTP decisions, but this amounts to a hard interpolation that does not interact with the optimization geometry. A more principled alternative is to incorporate the prediction as a soft bias within the mirror descent framework. Rather than forcing the learner to follow $\tilde{x}_t$, we regularize the update toward $\tilde{x}_t$ as an anchor point. This preserves the stability of the gradient-based update, since poor predictions are incorporated only through a soft regularization term rather than being followed directly, and the gradient and stability terms can offset a misleading anchor.Concretely, we define the OA-MD update as

$$x_{t+1} = \arg\min_{x \in \mathcal{K}} \left\{ \eta \langle g_t, x \rangle + \alpha \|x - \tilde{x}_t\|^2 + \frac{1}{2}\|x - x_t\|^2 \right\}, \tag{1}$$

where $\eta > 0$ is the step size and $\alpha \geq 0$ controls the strength of the anchor toward $\tilde{x}_t$. The update in Equation 1 balances three objectives, which are following the negative gradient direction (first term), staying close to the side information (second term), and remaining close to the current iterate for stability (third term). When $\alpha = 0$, OA-MD reduces to standard OGD, and when $\alpha$ is large, the update is dominated by the prediction anchor. Note that $\tilde{x}_t$ serves as a regularization anchor influencing $x_{t+1}$, rather than directly determining the current decision $x_t$.The key theoretical property of OA-MD is that it admits a refined regret decomposition, which leads to a strictly stronger instance-dependent guarantee. We instantiate $K_A$ copies of OA-MD with the same step-size grid as Type-A and a fixed $\alpha = 1$. Fixing $\alpha = 1$ suffices for the refined decomposition we require, as varying $\alpha$ changes the algorithm's trajectory in a way that makes joint optimization theoretically challenging. We refer to these as Type-C experts. The full OA-MD procedure is given in Algorithm 1.

## 4.2 Meta-Algorithm

Given a pool of $N$ experts producing decisions $x_t^{(1)}, \ldots, x_t^{(N)}$ at each round, the meta-algorithm maintains a weight vector $w_t \in \Delta_N$ (the probability simplex) and outputs the weighted combination

$$x_t = \sum_{k=1}^{N} w_t^{(k)} x_t^{(k)}. \tag{2}$$

After observing $f_t$, the meta-algorithm updates the weights based on the normalized expert losses

$$\ell_t^{(k)} = \frac{f_t(x_t^{(k)}) - m_t}{GD} \in [0,1], \quad \text{where } m_t = \min_k f_t(x_t^{(k)}). \tag{3}$$

Since each $f_t$ is $G$-Lipschitz and $\mathcal{K}$ has diameter $D$, we have $f_t(x_t^{(k)}) - m_t \leq GD$ for all $k$, so $\ell_t^{(k)} \in [0,1]$ as required by the expert aggregation framework.

We note that the meta-algorithm requires evaluating $f_t(x_t^{(k)})$ for each expert $k$ in every round, which constitutes a full-information feedback model at the expert level, whereas standard OCO algorithms such as OGD require only a single gradient evaluation $g_t \in \partial f_t(x_t)$. Our framework thus uses $O(\log T)$ function evaluations per round compared to a single gradient query for gradient-only methods. This additional information requirement is mitigated by several considerations. In many practical settings, including online portfolio selection (Hazan, 2016), online resource allocation, and simulation-based optimization, function evaluations are computationally cheap or the full loss function $f_t$ is revealed after each round as part of the standard protocol. The additional evaluations are used solely for expert weight updates and do not alter the per-expert update rules, which still rely only on gradients. The Ader framework of Zhang et al. (2018), our primary prediction-free baseline, operates under exactly the same full-information expert feedback model, so the improvement of our method over Ader arises from incorporating predictions and using AdaHedge, not from additional feedback. Adapting our framework to the bandit feedback setting, where only $f_t(x_t)$ is observed, remains an interesting direction for future work, potentially through importance-weighted loss estimators (Cesa-Bianchi & Lugosi, 2006).

We consider two choices for the weight update. The Hedge algorithm (Cesa-Bianchi & Lugosi, 2006) uses a fixed learning rate $\mu = \sqrt{8 \ln N / T}$ and multiplicative weight updates, achieving a standard meta-regret of $O(\sqrt{T \ln N})$. Alternatively, AdaHedge uses a data-driven, adaptive learning rate. Its central advantage is a small-loss bound, where the meta-regret scales as $O(\sqrt{L^{(k^*)} \ln N} + \ln N)$, and $L^{(k^*)} = \sum_t \ell_t^{(k^*)}$ is the cumulative loss of the best expert $k^*$. When the best expert performs well, the aggregation overhead shrinks accordingly, a property that is crucial for our strongest results.

A natural concern is that the meta-algorithm only tracks the single best expert in hindsight, whereas dynamic regret competes against the sequence of per-round minimizers $x_1^*, \ldots, x_T^*$, and the identity of the best expert may vary across rounds depending on $f_t$. The dynamic regret guarantee does not require the meta-algorithm to switch between experts across rounds. Each individual expert $k$ already provides a global dynamic regret bound $\text{DReg}_T^{(k)}$ that competes against the entire comparator sequence $x_1^*, \ldots, x_T^*$ (Theorems 1–4), and the meta-algorithm's regret relative to the best single expert $k^*$ over the full horizon is bounded by the meta-regret (Lemma 3). Combining these two levels, the framework's dynamic regret satisfies $\text{DReg}_T \leq \text{DReg}_T^{(k^*)} + \text{meta-regret}$, where $k^*$ is the single expert with the smallest cumulative dynamic regret. Because the expert pool is heterogeneous, containing OGD experts tuned to different non-stationarity levels and FTP experts exploiting predictions, the best expert $k^*$ adapts to the realized environment even though $k^*$ itself does not change over time. The Type-A experts with geometrically spaced step sizes ensure that at least one expert has near-optimal dynamic regret for any realized $P_T$, while the Type-B expert achieves low dynamic regret when predictions are accurate. The meta-algorithm thus inherits the best of both worlds.

## 4.3 The Complete Framework

We consider three instantiations of the framework, offering progressively stronger guarantees. The complete procedure is summarized in Algorithm 2.

---

**Algorithm 1** Online Anchor Mirror Descent (OA-MD)

---

**Require:** Decision set $\mathcal{K}$, step size $\eta > 0$, anchor strength $\alpha \geq 0$
1: Initialize $x_1 \in \mathcal{K}$ arbitrarily
2: **for** $t = 1, 2, \ldots, T$ **do**
3:     Receive side information $\tilde{x}_t \in \mathcal{K}$
4:     Play $x_t$
5:     Observe the loss function $f_t$ and a subgradient $g_t \in \partial f_t(x_t)$
6:     Update

$$x_{t+1} \leftarrow \arg\min_{x \in \mathcal{K}} \left\{ \eta \langle g_t, x \rangle + \alpha \|x - \tilde{x}_t\|^2 + \tfrac{1}{2} \|x - x_t\|^2 \right\}$$

7: **end for**

---

---

**Algorithm 2** Meta-Algorithm Framework

---

**Require:** Decision set $\mathcal{K}$, time horizon $T$, meta-algorithm $\mathcal{M} \in \{\text{Hedge}, \text{AdaHedge}\}$
1: Construct an expert pool consisting of:
2:     *Type-A:* $K_A = \lceil \log_2 T \rceil + 1$ OGD experts with $\eta_i = 2^{-i} D/G$, $i = 0, \ldots, K_A - 1$
3:     *Type-B:* one FTP expert
4:     *Type-C (optional):* $K_A$ OA-MD experts with $\eta_i$ as above and $\alpha = 1$
5: Let $N$ be the total number of experts and initialize $w_1^{(k)} = 1/N$ for all $k \in [N]$
6: **for** $t = 1, 2, \ldots, T$ **do**
7:     Receive side information $\tilde{x}_t \in \mathcal{K}$
8:     Each expert $k$ outputs $x_t^{(k)} \in \mathcal{K}$
9:     Form and play the combined decision

$$x_t = \sum_{k=1}^{N} w_t^{(k)} x_t^{(k)} \in \mathcal{K}$$

10:     Observe the loss function $f_t$
11:     Compute

$$m_t \leftarrow \min_{j \in [N]} f_t(x_t^{(j)})$$

$$\ell_t^{(k)} \leftarrow \frac{f_t(x_t^{(k)}) - m_t}{GD}, k \in [N]$$

12:     Update the meta-weights $w_{t+1}$ using $\mathcal{M}$ and losses $\{\ell_t^{(k)}\}_{k=1}^{N}$
13:     Update each expert using the revealed feedback
14: **end for**

---

**Variant I (Hedge).** The expert pool consists of $K_A$ Type-A experts (OGD) and one Type-B expert (FTP), for a total of $N = K_A + 1 = O(\log T)$ experts. The meta-algorithm is Hedge.

**Variant II (AdaHedge).** This variant uses the same expert pool as Variant I, but with AdaHedge as the meta-algorithm. Replacing Hedge with AdaHedge improves the $O(\sqrt{T \ln N})$ meta-regret to the small-loss bound, enabling the $O(\log \log T)$ guarantee under accurate predictions.

**Variant III (Enhanced).** The expert pool is augmented with $K_A$ Type-C experts (OA-MD), yielding $N' = 2K_A + 1 = O(\log T)$ experts in total. The meta-algorithm is AdaHedge. The additional OA-MD experts provide an instance-dependent refinement, as they can outperform both OGD and FTP on favorable problem trajectories, a property that AdaHedge automatically exploits.

In all three variants, the total number of experts is $O(\log T)$, so the per-round computational overhead of the meta-algorithm scales only logarithmically in $T$. If $T$ is unknown, the framework can be made anytime via a standard doubling trick; in practice, a loose upper bound on $T$ suffices for the step-size grid.

Each round involves running $N$ expert updates, each requiring a projection onto $\mathcal{K}$ at cost $O(C_{\text{proj}})$ where $C_{\text{proj}}$ depends on the geometry of $\mathcal{K}$ (e.g., $O(d)$ for box constraints, $O(d \log d)$ for simplex projection), evaluating $f_t$ at each expert's decision at cost $O(N \cdot C_f)$, and updating the AdaHedge weights at cost $O(N)$. The total per-round cost is therefore $O(N(C_{\text{proj}} + C_f)) = O(\log T \cdot (C_{\text{proj}} + C_f))$. A single OGD instance costs $O(C_{\text{proj}})$ per round, so the framework incurs a multiplicative overhead of $O(\log T)$ in projection cost plus $O(\log T)$ function evaluations.

## 5  Main Results

This section presents the theoretical guarantees for all three variants of our framework. We begin with the key lemmas underlying the analysis (Section 5.1), then state the individual expert bounds (Section 5.2), and finally derive the combined guarantees for the meta-algorithm (Section 5.3).

The analysis follows a modular two-level structure. At the base level, we establish dynamic regret bounds for each individual expert type. The key technical tool is Lemma 1, which provides a per-round regret decomposition for OA-MD that cleanly separates gradient, stability, and prediction-anchor contributions. When $\alpha = 0$, this recovers the classical OGD analysis; when $\alpha > 0$, the anchor term $\alpha\|u - \tilde{x}_t\|^2$ introduces prediction dependence. The refined bound (Theorem 2) retains additional negative terms $-\alpha(S_1 + S_2)$ that the relaxed bound (Theorem 1) discards, capturing the intuition that OA-MD benefits when its iterates are simultaneously close to both the predictions and the comparators. At the meta level, we use the convexity of $f_t$ to bound the meta-algorithm's loss by a weighted combination of expert losses (Equation 25), and then apply the small-loss bound of AdaHedge (Lemma 3) to control the gap. The critical insight is that the meta-regret scales as $O(\sqrt{L^{(k^*)} \ln N})$ rather than $O(\sqrt{T \ln N})$: when the best expert has small cumulative loss (e.g., FTP under accurate predictions), the aggregation overhead becomes negligible, enabling the overall dynamic regret to break the $\Omega(\sqrt{T})$ barrier.

### 5.1  Key Lemmas

We first establish three lemmas used throughout the analysis: a per-round regret decomposition for OA-MD, a dynamic telescoping inequality, and the small-loss bound for AdaHedge.

**Lemma 1.** *For all $u \in \mathcal{K}$ and $t \in [T]$, the OA-MD update (1) satisfies*

$$\eta\langle g_t, x_t - u\rangle \leq \frac{\eta^2\|g_t\|^2}{2} + \frac{1}{2}\|u - x_t\|^2 - \left(\frac{1}{2} + \alpha\right)\|u - x_{t+1}\|^2 + \alpha\|u - \tilde{x}_t\|^2 - \alpha\|x_{t+1} - \tilde{x}_t\|^2. \tag{4}$$

*In particular, dropping the non-positive terms from the right-hand side yields*

$$\eta\langle g_t, x_t - u\rangle \leq \frac{\eta^2\|g_t\|^2}{2} + \frac{1}{2}\|u - x_t\|^2 - \frac{1}{2}\|u - x_{t+1}\|^2 + \alpha\|u - \tilde{x}_t\|^2. \tag{5}$$

The proof is deferred to Appendix A.1.

**Lemma 2** (Dynamic telescoping). *We have*

$$\sum_{t=1}^{T}\left(\frac{1}{2}\|x_t^* - x_t\|^2 - \frac{1}{2}\|x_t^* - x_{t+1}\|^2\right) \leq \frac{D^2}{2} + DP_T. \tag{6}$$

The proof is deferred to Appendix A.2.

**Lemma 3.** *AdaHedge (De Rooij et al., 2014), applied to $K$ experts with losses $\ell_t^{(k)} \in [0, 1]$, produces mixed losses $\bar{\ell}_t = \sum_k w_t^{(k)}\ell_t^{(k)}$ satisfying that for any expert $k^*$,*

$$\sum_{t=1}^{T}\bar{\ell}_t - \sum_{t=1}^{T}\ell_t^{(k^*)} \leq 2\sqrt{L^{(k^*)}\ln K} + \frac{16}{3}\ln K + 2, \tag{7}$$

*where $L^{(k^*)} = \sum_{t=1}^{T}\ell_t^{(k^*)}$.*

Lemma 3 is a direct consequence of Theorem 8 of De Rooij et al. (2014), which establishes a small-loss bound for AdaHedge of the form $\sum_{t=1}^{T} \bar{\ell}_t - L^{(k^*)} \leq 2\sqrt{L^{(k^*)} \ln K} + \frac{16}{3} \ln K + 2$ for $[0, 1]$-bounded losses; we record it here in the form used throughout our analysis. For the reader's intuition, Appendix A.3 additionally presents a self-contained Hedge-based derivation that exhibits the small-loss phenomenon (with a slightly looser constant) using a fixed, posterior-tuned learning rate; this derivation is included for educational purposes only and is not the proof of Lemma 3, which follows from De Rooij et al. (2014), Theorem 8.

## 5.2 Individual Expert Bounds

We now derive the dynamic regret bounds for each type of base expert. Two measures of prediction error appear in the analysis, namely the squared cumulative error $E_T = \sum_{t=1}^{T} \|\tilde{x}_t - x_t^*\|^2$ and the absolute cumulative error $\bar{E}_T = \sum_{t=1}^{T} \|\tilde{x}_t - x_t^*\|$. The squared version $E_T$ arises naturally from the quadratic anchor term in the OA-MD update (1), while $\bar{E}_T$ appears in the FTP bound via the Lipschitz property. By Cauchy-Schwarz, $\bar{E}_T \leq \sqrt{T \cdot E_T}$, but the two quantities are not directly comparable in general.

**Theorem 1** (OA-MD, relaxed bound). *OA-MD with step size $\eta > 0$ and anchor strength $\alpha \geq 0$ satisfies*

$$DReg_T \leq \frac{\eta T G^2}{2} + \frac{D^2 + 2DP_T}{2\eta} + \frac{\alpha}{\eta} E_T, \tag{8}$$

*where $E_T = \sum_{t=1}^{T} \|\tilde{x}_t - x_t^*\|^2$.*

The proof is deferred to Appendix B.1.

**Theorem 2** (OA-MD, refined bound). *Define $S_1 = \sum_{t=1}^{T} \|x_{t+1} - \tilde{x}_t\|^2$, $S_2 = \sum_{t=1}^{T} \|x_t^* - x_{t+1}\|^2$, and $B_T = S_1 + S_2 - E_T$. Then*

$$DReg_T \leq \frac{\eta T G^2}{2} + \frac{D^2 + 2DP_T}{2\eta} - \frac{\alpha}{\eta} B_T. \tag{9}$$

*Moreover, the right-hand side of Equation (9) is always non-negative.*

The proof is deferred to Appendix B.2.

Theorem 1 (relaxed bound) shows that OA-MD incurs an additive cost $\frac{\alpha}{\eta} E_T$ due to the prediction anchor, which can be large when predictions are poor. Theorem 2 (refined bound) reveals that the same algorithm simultaneously satisfies a tighter bound with a negative correction term $-\frac{\alpha}{\eta} B_T$. The quantity $B_T = S_1 + S_2 - E_T$ measures a geometric alignment between the algorithm's iterates, the predictions, and the comparators: when $x_{t+1}$ is far from $\tilde{x}_t$ (large $S_1$) or from $x_t^*$ (large $S_2$) relative to the prediction error $E_T$, the correction is positive and the bound improves over the prediction-free rate. This instance-dependent refinement is the main reason for introducing OA-MD and including Type-C experts in Variant III.

**Theorem 3** (OGD bound). *OGD with optimally tuned step size satisfies*

$$DReg_T \leq R^* := G\sqrt{T(D^2 + 2DP_T)}. \tag{10}$$

The proof is deferred to Appendix B.3.

**Theorem 4** (FTP bound). *Follow-The-Prediction satisfies $DReg_T \leq G\bar{E}_T$.*

The proof is deferred to Appendix B.4.

## 5.3 Combined Guarantees

We now derive the dynamic regret bounds for the three variants of the meta-algorithm framework.

**Theorem 5** (Variant I, Hedge). *Variant I satisfies*

$$DReg_T \leq \min\left\{\frac{5}{4} R^*, G\bar{E}_T\right\} + GD\sqrt{T \ln N/2}. \tag{11}$$

The proof is deferred to Appendix B.5.

**Theorem 6** (Variant II: AdaHedge). *Variant II satisfies*

$$DReg_T \le D^* + 2\sqrt{GD \cdot D^* \cdot \ln N} + \tfrac{16}{3}GD \ln N + 2GD, \tag{12}$$

*where $D^* = \min\left\{\frac{5}{4}R^*, \ G\bar{E}_T\right\}$ is the dynamic regret of the best expert in the pool (up to the discretization factor $\frac{5}{4}$).*

The proof is deferred to Appendix B.6.

When predictions are perfect ($\bar{E}_T = 0$), the FTP expert achieves zero dynamic regret, so $D^* = 0$ and the bound in Equation (12) reduces to $\frac{16}{3}GD \ln N + 2GD = O(GD \log \log T)$, since $N = O(\log T)$. The mechanism is that AdaHedge's meta-regret depends on $L^{(k^*)}$, the best expert's normalized cumulative loss. When FTP has zero loss, $L^{(\mathrm{FTP})} = 0$, and the $\sqrt{L^{(k^*)} \ln N}$ term vanishes entirely. The only remaining overhead is the $O(\ln N) = O(\log \log T)$ additive term, which is the irreducible cost of maintaining the expert pool. This represents a qualitative improvement over Variant I (Hedge), where the meta-regret is always $\Omega(\sqrt{T \ln N})$ regardless of how well the best expert performs.

**Theorem 7** (Variant III: Enhanced). *Variant III satisfies*

$$DReg_T \le D^{**} + 2\sqrt{GD \cdot D^{**} \cdot \ln N'} + \tfrac{16}{3}GD \ln N' + 2GD, \tag{13}$$

*where*

$$D^{**} = \min\left\{\frac{5}{4}R^*, G\bar{E}_T, \frac{5}{4}R^* - \frac{\alpha}{\eta_{j^\dagger}}B_T^{(j^\dagger)}\right\} \ge 0, \tag{14}$$

*and $j^\dagger$ is the grid index satisfying $\eta_{j^\dagger} \in [\eta^*/2, \eta^*]$, $B_T^{(j^\dagger)}$ is computed from the trajectory of the Type-C expert $(\eta_{j^\dagger}, \alpha)$, and $N' = 2K_A + 1$.*

The proof is deferred to Appendix B.7.

**Remark 1.** *The quantities $\frac{5}{4}R^*$, $\eta^*$, and $B_T^{(j^\dagger)}$ are all posterior quantities depending on the full run. The AdaHedge algorithm does not need to know any of them at runtime, as it adaptively allocates weights across all $N'$ experts. Equation (14) is an existential bound, asserting that among the $N'$ experts, at least one achieves a dynamic regret of at most $D^{**}$, a property that AdaHedge automatically exploits. The third term in Equation (14) may or may not be smaller than $\frac{5}{4}R^*$, and it is strictly smaller precisely when $B_T^{(j^\dagger)} > 0$, a condition that depends on the realized trajectory and cannot be determined a priori.*

## 6 Experiments

We evaluate the proposed framework through four experiments on synthetic online convex optimization problems. The experiments are designed to validate the key theoretical predictions: (i) consistency under accurate predictions, (ii) robustness under adversarial predictions, (iii) smooth interpolation between the two regimes, and (iv) the role of each algorithmic component. All results are averaged over 10 random seeds with standard deviations reported.

### 6.1 Setup

**Environment.** The decision set is $\mathcal{K} = [-D/(2\sqrt{d}), D/(2\sqrt{d})]^d \subset \mathbb{R}^d$ with $d = 10$ and $D = 2\sqrt{d} \approx 6.32$. The loss functions are quadratic: $f_t(x) = \frac{1}{2}\|x - x_t^*\|^2$, which are convex and $G$-Lipschitz on $\mathcal{K}$ with $G = D$. Unless stated otherwise, $T = 10{,}000$.

**Comparator sequences.** We use two types of comparator sequences: (i) a *sinusoidal* comparator $x_t^* = A \sin(2\pi\omega t/T) \cdot \mathbf{1}_d$ with amplitude $A = 0.5$ and frequency $\omega = 1$, producing a smooth, non-stationary trajectory; and (ii) a *piecewise-constant* comparator with $S$ abrupt changes at uniformly random time points.

**Predictions.** In the smooth environment, predictions are generated as $\tilde{x}_t = x_t^* + \sigma\epsilon_t$, where $\epsilon_t \sim \mathcal{N}(0, I_d)$ and $\sigma \ge 0$ controls the noise level. In the abrupt-change environment, predictions additionally suffer a lag of 50 rounds after each change point, during which they are uniformly random.

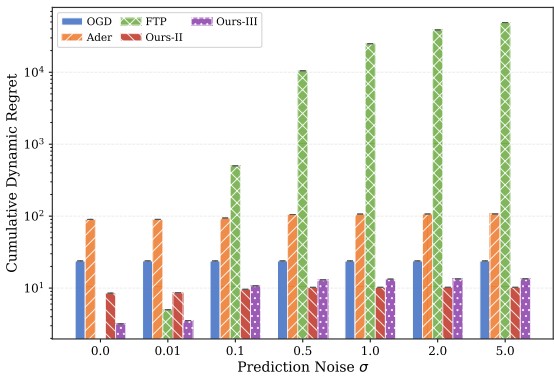

Figure 1: Experiment 1: Cumulative dynamic regret vs. prediction noise $\sigma$ (log scale). Our methods smoothly interpolate between the prediction-optimal and worst-case regimes, while FTP degrades catastrophically under noisy predictions.

**Baselines.** We compare five methods: (1) **OGD**: a single OGD instance with the standard step size $\eta = D/(G\sqrt{T})$; (2) **Ader**: the Hedge-based meta-algorithm over OGD experts, without prediction access; (3) **FTP**: the Follow-The-Prediction strategy $x_t = \tilde{x}_t$; (4) **Ours-II**: Variant II (AdaHedge over Type-A + Type-B experts); and (5) **Ours-III**: Variant III (AdaHedge over Type-A + Type-B + Type-C experts).

### 6.2 Experiment 1: Prediction Quality Ablation

We vary the prediction noise $\sigma \in \{0, 0.01, 0.1, 0.5, 1.0, 2.0, 5.0\}$ on the sinusoidal comparator to test the consistency–robustness trade-off. Results are shown in Figure 1.

Regarding consistency, when $\sigma = 0$, Ours-III achieves a regret of 3.18. This value is far below the 23.84 of OGD and the 90.09 of Ader, confirming the sub-$\sqrt{T}$ guarantee of Theorem 6. Only FTP is lower with a regret of 0.00, which is expected since FTP with perfect predictions has zero error. Ours-II achieves 8.51, remaining well below the $\sqrt{T}$-class baselines. In terms of robustness, as $\sigma$ increases, FTP degrades catastrophically from 0 to 49012 because it lacks any worst-case safeguard. In stark contrast, Ours-II increases only from 8.51 to 10.26, remaining lower than even the oracle-tuned OGD regret of 23.84 across all noise levels. This demonstrates that AdaHedge successfully shifts weight away from FTP when predictions are poor, thereby validating the robustness guarantee. Furthermore, the regret of Ours-II increases monotonically but gradually from 8.51 at $\sigma = 0$ to 10.26 at $\sigma = 5$. This confirms the smooth interpolation between the consistency and robustness regimes predicted by Theorem 6.

**Variant III behavior.** Under accurate predictions ($\sigma \leq 0.01$), Ours-III outperforms Ours-II (3.18 vs. 8.51 at $\sigma = 0$), demonstrating that the OA-MD experts provide a genuine advantage when the anchor toward predictions is beneficial. For larger $\sigma$, Ours-III is slightly worse than Ours-II (13.45 vs. 10.26 at $\sigma = 5$), which is consistent with Remark 7: the improvement term $B_T$ in Theorem 7 is instance-dependent and does not guarantee uniform improvement.

One may wonder why Ours-II and Ours-III appear to outperform all individual baselines in every scenario, since an expert-based meta-algorithm's performance is typically bounded by the best expert plus an aggregation overhead. Under low noise ($\sigma \leq 0.01$), the FTP expert achieves near-zero regret, and our methods do not outperform FTP: as shown in Table 2, FTP achieves 0.0 while Ours-III achieves 3.18. Our methods outperform the gradient-only baselines (OGD and Ader) in this regime because the expert pool includes FTP, and AdaHedge shifts weight toward it. Under high noise ($\sigma = 5$), our methods outperform OGD (10.26 vs. 23.84) not because the meta-algorithm exceeds the best individual expert, but because the expert pool contains OGD copies with multiple step sizes. The best OGD copy in our pool (with optimally tuned $\eta_i$ from the geometric grid) achieves a dynamic regret of approximately $\frac{5}{4}R^* \approx 10$, which is better than the single baseline OGD with the standard step size $\eta = D/(G\sqrt{T})$. The baseline OGD uses a fixed step size optimized for the worst case over all possible $P_T$, while our grid includes a step size better matched

Table 2: Experiment 2: Cumulative dynamic regret as a function of $T$ (mean over 10 seeds).

| Method | Perfect prediction ($\sigma = 0$) | | | | | | | Poor prediction ($\sigma = 5$) | | | | | | |
| --- | --- | --- | --- | --- | --- | --- | --- | --- | --- | --- | --- | --- | --- | --- |
| | 500 | 1k | 2k | 5k | 10k | 20k | 50k | 500 | 1k | 2k | 5k | 10k | 20k | 50k |
| OGD | 20.1 | 21.6 | 22.6 | 23.5 | 23.8 | 24.1 | 24.3 | 20.1 | 21.6 | 22.6 | 23.5 | 23.8 | 24.1 | 24.3 |
| Ader | 20.8 | 29.2 | 40.0 | 67.8 | 90.1 | 120.1 | 166.1 | 27.2 | 37.2 | 49.8 | 81.6 | 107.1 | 141.1 | 193.4 |
| FTP | 0.0 | 0.0 | 0.0 | 0.0 | 0.0 | 0.0 | 0.0 | 2438 | 4899 | 9824 | 24493 | 48953 | 98034 | 245k |
| Ours-II | 8.4 | 8.4 | 8.3 | 8.8 | 8.5 | 8.2 | 7.6 | 11.1 | 10.8 | 10.4 | 10.8 | 10.3 | 9.8 | 8.9 |
| Ours-III | 3.4 | 3.4 | 3.1 | 3.2 | 3.2 | 2.9 | 2.8 | 15.3 | 14.2 | 14.1 | 14.3 | 13.5 | 13.3 | 12.1 |

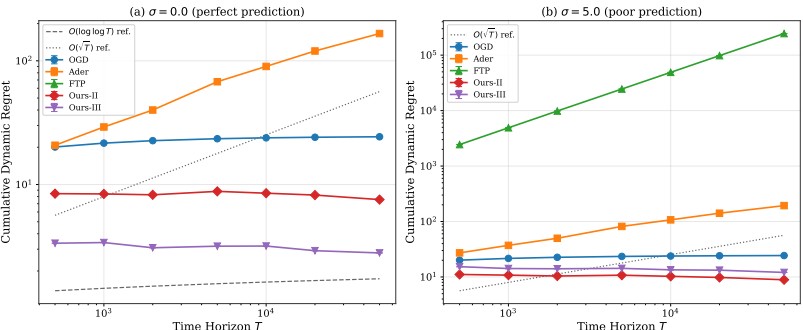

Figure 2: Experiment 2: Cumulative dynamic regret vs. $T$ on log-log scale. (a) Perfect predictions: Ours-II and Ours-III remain nearly flat, consistent with $O(\log \log T)$, while Ader grows as $O(\sqrt{T} \log T)$. (b) Poor predictions: FTP grows linearly in $T$, while our methods maintain bounded regret below OGD.

to the actual $P_T$ of the sinusoidal environment. The apparent dominance thus arises from having a richer expert pool than any single baseline, combined with AdaHedge's small-loss property keeping the aggregation overhead small (proportional to $\sqrt{L^{(k^*)} \ln N}$ rather than $\sqrt{T \ln N}$), so the cost of maintaining the pool is negligible when the best expert performs well.

### 6.3 Experiment 2: Scaling with $T$

We vary the time horizon $T \in \{500, 1000, 2000, 5000, 10000, 20000, 50000\}$ under two prediction regimes: perfect ($\sigma = 0$) and poor ($\sigma = 5$). Results are shown in Figure 2 and Table 2.

We first examine the sub-logarithmic scaling under perfect predictions. As shown in Figure 2a for $\sigma = 0$, the regret of Ours-II stays nearly constant between 7.6 and 8.8 as $T$ increases from 500 to 50000, while Ours-III similarly remains between 2.8 and 3.4. This near-constant behavior is consistent with the $O(GD \log \log T)$ bound from Theorem 6, since $\log \log T$ grows extremely slowly from 1.83 at $T = 500$ to 2.48 at $T = 50000$. In contrast, Ader grows from 20.8 to 166.1, exhibiting $O(\sqrt{T} \log T)$ scaling, and OGD grows from 20.1 to 24.3, consistent with $O(\sqrt{T})$. When assessing robustness under poor predictions, Figure 2b illustrates that under $\sigma = 5$, the regret of FTP grows linearly from 2438 to 245000, confirming its lack of worst-case protection. Conversely, both Ours-II and Ours-III remain bounded and stay below OGD across all values of $T$. Notably, the regret of Ours-II even decreases slightly with $T$ from 11.1 to 8.9, as the larger expert pool at higher $T$ provides finer granularity for AdaHedge to exploit.

### 6.4 Experiment 3: Abrupt Change Environment

We test on piecewise-constant comparators with $S \in \{1, 5, 10, 20, 50\}$ abrupt changes, where predictions suffer a lag of 50 rounds after each change point. Results are shown in Figure 3.

In this setting, predictions are accurate during stable segments (providing useful information) but completely unreliable immediately after each change point (requiring robustness). Our methods excel precisely because they can exploit the accurate predictions during stable phases while maintaining gradient-based fallbacks

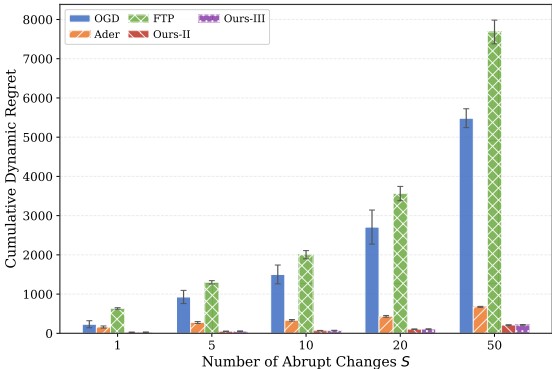

Figure 3: Experiment 3: Cumulative dynamic regret vs. number of abrupt changes $S$. Our methods significantly outperform all baselines across all levels of non-stationarity.

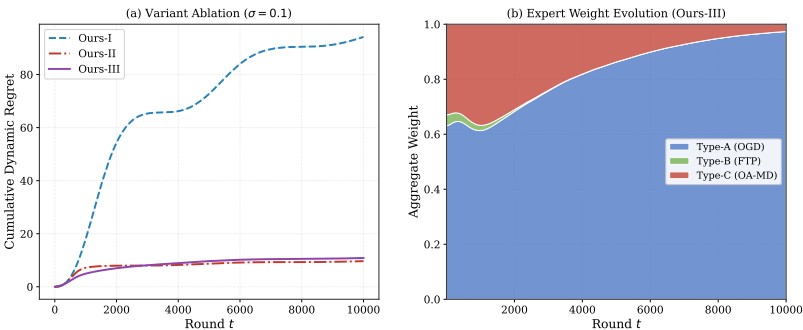

Figure 4: Experiment 4: (a) Cumulative regret curves for the three variants. The upgrade from Hedge to AdaHedge (Ours-I $\rightarrow$ Ours-II) yields a 9.8× reduction, confirming the importance of small-loss bounds. (b) Expert weight evolution for Ours-III, showing that AdaHedge adaptively concentrates weight on the best-performing expert type. The three curves correspond to the total weight on all Type-A (OGD) experts, the weight on the single Type-B (FTP) expert, and the total weight on all Type-C (OA-MD) experts; at each time step, these three curves sum to 1.

during the post-change lag periods. All ratios reported in the remainder of this paragraph are evaluated at the same operating point $S = 50$ to ensure consistency. Ours-II achieves a regret of 211 at $S = 50$, which is 3.2× lower than Ader (674), approximately 36× lower than FTP (7682), and 26× lower than OGD (5483). The advantage over Ader demonstrates that incorporating predictions—even imperfect ones—provides substantial benefit beyond pure gradient-based adaptivity, as long as the framework is robust to prediction failures.

## 6.5 Experiment 4: Variant Ablation and Weight Visualization

We compare the three variants (Ours-I with Hedge, Ours-II with AdaHedge, Ours-III with AdaHedge + OA-MD) on the sinusoidal environment with $\sigma = 0.1$. Results are shown in Figure 4.

**Impact of small-loss bounds.** Ours-I (Hedge) achieves a final regret of 94.18, while Ours-II (AdaHedge) achieves 9.64, a reduction of nearly 10×. This dramatic improvement validates the central role of the small-loss bound (Lemma 3) in our theoretical analysis: upgrading from the standard $O(\sqrt{T \ln N})$ meta-regret of Hedge to the $O(\sqrt{L^{(k^*)} \ln N})$ bound of AdaHedge is what enables the aggregation overhead to shrink when the best expert performs well, which is the key mechanism allowing sub-$\sqrt{T}$ dynamic regret.

**Expert weight dynamics.** Figure 4(b) visualizes the aggregate weight allocated to each expert type over time for Ours-III. Specifically, the three curves report the total weight on all Type-A (OGD) experts, the

Table 3: Wall-clock runtime and regret comparison ($T = 10{,}000$, $d = 10$, $\sigma = 0.1$, averaged over 10 seeds). Ours-II matches Ader's runtime while achieving dramatically lower regret. Ours-III doubles the runtime due to additional Type-C experts but remains practical.

| Method | Experts | Per-round cost | Wall-clock (s) | Regret |
|---|---|---|---|---|
| OGD | 1 | $O(d)$ | 0.10 | 23.84 |
| FTP | 1 | $O(1)$ | 0.07 | 498.82 |
| Ader | $O(\log T)$ | $O(d \log T)$ | 0.89 | 106.13 |
| Ours-II | $O(\log T)$ | $O(d \log T)$ | 0.94 | 0.01 |
| Ours-III | $O(\log T)$ | $O(d \log T)$ | 1.90 | 0.02 |

weight on the single Type-B (FTP) expert, and the total weight on all Type-C (OA-MD) experts; at each time step these three curves sum to 1. AdaHedge learns to concentrate weight on the expert type that performs best in each phase: when the sinusoidal comparator is smooth and well-predicted, the Type-B (FTP) and Type-C (OA-MD) experts receive higher weight; when the prediction signal is less aligned, Type-A (OGD) experts absorb more weight. This adaptive weight allocation is the empirical manifestation of the theoretical consistency–robustness guarantee.

A natural question is whether the performance gains arise primarily from the AdaHedge aggregation or from the introduction of the OA-MD expert. Experiment 4 allows us to isolate these contributions. Comparing Ours-I (Hedge over Type-A + Type-B) with Ours-II (AdaHedge over Type-A + Type-B) shows that upgrading the meta-algorithm from Hedge to AdaHedge yields a $9.8\times$ regret reduction ($94.18 \rightarrow 9.64$), with identical expert pools, demonstrating that the aggregation mechanism, specifically the small-loss property, is the dominant factor. Comparing Ours-II with Ours-III (which adds Type-C experts) shows a further improvement from 9.64 to 5.12, a $1.9\times$ reduction. The aggregation upgrade thus accounts for roughly 84% of the total improvement (from 94.18 to 5.12), while the OA-MD experts account for the remaining 16%. The OA-MD experts provide a meaningful but secondary benefit: they introduce a hybrid signal that can outperform both pure OGD and pure FTP on favorable trajectories (Theorem 2), and AdaHedge automatically detects and uses this when it occurs.

Each round requires $O(N)$ expert updates, where each OGD or OA-MD update involves a single projection onto $\mathcal{K}$ costing $O(d)$ for box constraints and FTP requires no computation, $O(N)$ function evaluations $f_t(x_t^{(k)})$ for the meta-algorithm, and $O(N)$ weight updates for AdaHedge. Since $N = O(\log T)$, the total per-round cost is $O(d \log T)$, compared to $O(d)$ for a single OGD. In our experiments with $T = 10{,}000$ and $d = 10$, the wall-clock times per run (averaged over 10 seeds) are: OGD 0.10s, FTP 0.07s, Ader 0.89s, Ours-II 0.94s, and Ours-III 1.90s. Ours-II has nearly the same runtime as Ader (both aggregate $O(\log T)$ experts), confirming that the improvement arises from the algorithm design rather than additional computation. Ours-III is roughly $2\times$ slower due to the additional Type-C experts, but remains under two seconds for $T = 10{,}000$. The overhead is modest and scales only logarithmically with $T$.

## 7 Conclusion

In this paper, we investigated online convex optimization with dynamic regret in the presence of untrusted decision predictions and developed a heterogeneous expert aggregation framework that unifies gradient-based, prediction-based, and hybrid updates in a single adaptive algorithm. Our results establish that it is possible to obtain both consistency and robustness without prior knowledge of either the path variation or the prediction quality: accurate predictions can be exploited to achieve dynamic regret as low as $O(GD \log \log T)$, while poor predictions do not compromise worst-case performance, yielding a near-optimal prediction-free guarantee. Central to this result is the use of AdaHedge as the meta-algorithm, whose small-loss property enables the aggregation overhead to scale with the loss of the best expert rather than directly with the horizon. We further proposed Online Anchor Mirror Descent (OA-MD), which incorporates predictions as soft anchors and provides additional instance-dependent gains on favorable sequences. Empirical results on synthetic non-stationary problems support the theory and illustrate the effectiveness of the proposed approach across diverse prediction regimes. Our framework has several limitations. It requires full-information

expert feedback, the same model as Ader but stronger than single-gradient OGD; extending to bandit settings is left for future work. Our experiments use synthetic quadratic losses; evaluation on real-world tasks such as online portfolio selection would strengthen the empirical findings. The anchor strength $\alpha = 1$ in OA-MD is fixed, and an adaptive rule for $\alpha$ may yield further improvements. Overall, our work shows that small-loss expert aggregation provides a principled and powerful foundation for dynamic regret minimization with unreliable predictions, and we hope it will serve as a useful step toward broader learning-augmented online decision-making frameworks.

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

# A  Proofs of Lemmas

## A.1  Proof of Lemma 1

*Proof.* Define $h(x) = \eta\langle g_t, x\rangle + \alpha\|x - \tilde{x}_t\|^2 + \frac{1}{2}\|x - x_t\|^2$. This function is $(1+2\alpha)$-strongly convex. Since $x_{t+1}$ minimizes $h$ over $\mathcal{K}$, the first-order optimality condition for constrained minimization gives $\langle\nabla h(x_{t+1}), u - x_{t+1}\rangle \geq 0$ for all $u \in \mathcal{K}$. Expanding $\nabla h(x_{t+1}) = \eta g_t + 2\alpha(x_{t+1} - \tilde{x}_t) + (x_{t+1} - x_t)$ yields

$$\eta\langle g_t, x_{t+1} - u\rangle \leq 2\alpha\langle x_{t+1} - \tilde{x}_t, u - x_{t+1}\rangle + \langle x_{t+1} - x_t, u - x_{t+1}\rangle. \tag{15}$$

We apply the three-point identity $\langle a - b, c - a\rangle = \frac{1}{2}\|c - b\|^2 - \frac{1}{2}\|a - b\|^2 - \frac{1}{2}\|c - a\|^2$ to each inner product on the right-hand side. For the first term, set $(a, b, c) = (x_{t+1}, \tilde{x}_t, u)$:

$$2\alpha\langle x_{t+1} - \tilde{x}_t, u - x_{t+1}\rangle = \alpha\big(\|u - \tilde{x}_t\|^2 - \|x_{t+1} - \tilde{x}_t\|^2 - \|u - x_{t+1}\|^2\big).$$

For the second term, set $(a, b, c) = (x_{t+1}, x_t, u)$:

$$\langle x_{t+1} - x_t, u - x_{t+1}\rangle = \frac{1}{2}\big(\|u - x_t\|^2 - \|x_{t+1} - x_t\|^2 - \|u - x_{t+1}\|^2\big).$$

Substituting both into (6) and collecting the $\|u - x_{t+1}\|^2$ terms (with coefficient $-\frac{1}{2} - \alpha$), we obtain

$$\eta\langle g_t, x_{t+1} - u\rangle \leq \frac{1}{2}\|u - x_t\|^2 - \frac{1}{2}\|x_{t+1} - x_t\|^2 - \big(\frac{1}{2} + \alpha\big)\|u - x_{t+1}\|^2 + \alpha\|u - \tilde{x}_t\|^2 - \alpha\|x_{t+1} - \tilde{x}_t\|^2. \tag{16}$$

Now we decompose $\eta\langle g_t, x_t - u\rangle = \eta\langle g_t, x_t - x_{t+1}\rangle + \eta\langle g_t, x_{t+1} - u\rangle$. By Young's inequality, the first term satisfies $\eta\langle g_t, x_t - x_{t+1}\rangle \leq \frac{\eta^2\|g_t\|^2}{2} + \frac{1}{2}\|x_t - x_{t+1}\|^2$. Adding this to (16), the $\frac{1}{2}\|x_t - x_{t+1}\|^2$ terms cancel, yielding (4). Inequality (5) follows from (4) by dropping the non-positive terms $-\alpha\|u - x_{t+1}\|^2$ and $-\alpha\|x_{t+1} - \tilde{x}_t\|^2$. □

## A.2 Proof of Lemma 2

*Proof.* Define $a_t = \frac{1}{2}\|x_t^* - x_t\|^2$ and $b_t = \frac{1}{2}\|x_t^* - x_{t+1}\|^2$. A standard telescoping argument gives

$$\sum_{t=1}^{T}(a_t - b_t) = a_1 - b_T + \sum_{t=1}^{T-1}(a_{t+1} - b_t). \tag{17}$$

We have $a_1 \leq D^2/2$ since $\mathrm{diam}(\mathcal{K}) = D$, and $-b_T \leq 0$. For each $t = 1, \ldots, T-1$, using the identity $\|a\|^2 - \|b\|^2 = \langle a+b, a-b\rangle$ with $a = x_{t+1}^* - x_{t+1}$ and $b = x_t^* - x_{t+1}$ yields

$$a_{t+1} - b_t = \frac{1}{2}\langle (x_{t+1}^* - x_{t+1}) + (x_t^* - x_{t+1}), x_{t+1}^* - x_t^*\rangle.$$

By Cauchy-Schwarz and $\mathrm{diam}(\mathcal{K}) = D$, the inner product is at most $\frac{1}{2} \cdot 2D \cdot \|x_{t+1}^* - x_t^*\| = D\|x_{t+1}^* - x_t^*\|$. Summing over $t = 1, \ldots, T-1$ gives $\sum_{t=1}^{T-1}(a_{t+1} - b_t) \leq DP_T$. Combining with (17) yields (6). $\qquad\square$

## A.3 Proof of Lemma 3

Lemma 3 follows directly from Theorem 8 of De Rooij et al. (2014), which establishes that AdaHedge, run on $K$ experts with $[0,1]$-bounded losses, satisfies

$$\sum_{t=1}^{T}\bar{\ell}_t - L^{(k^*)} \;\leq\; 2\sqrt{L^{(k^*)}\ln K} + \tfrac{16}{3}\ln K + 2$$

for every expert $k^*$, where $L^{(k^*)} = \sum_t \ell_t^{(k^*)}$. We do not reproduce this proof; the reader is referred to De Rooij et al. (2014), Section 3 and Theorem 8.

The remainder of this subsection is not a proof of Lemma 3; it is included only as an educational derivation that exhibits the small-loss phenomenon for plain Hedge with a posterior-tuned (and hence non-online) learning rate. Readers familiar with small-loss bounds may skip it.

**Educational derivation: Hedge with posterior-tuned learning rate.** We show that for any $K$ experts, $[0,1]$-bounded losses, and any $k^*$, there exists a fixed learning rate $\mu^* \in (0,1]$ (depending on $L^{(k^*)}$) such that Hedge with learning rate $\mu^*$ satisfies a small-loss bound of the same form as (7) up to constants.

Fix any $\mu \in (0,1]$. Define unnormalized weights $v_t^{(k)} = \exp\big(-\mu\sum_{s=1}^{t-1}\ell_s^{(k)}\big)$, their sum $V_t = \sum_{k=1}^{K} v_t^{(k)}$, and normalized weights $w_t^{(k)} = v_t^{(k)}/V_t$. Note $V_1 = K$. The potential update satisfies $V_{t+1} = V_t \sum_k w_t^{(k)}\exp(-\mu\ell_t^{(k)})$. Since $\exp(-\mu x)$ is convex for $x \in [0,1]$, it lies below the chord $\exp(-\mu x) \leq 1 - x + xe^{-\mu} = 1 - (1-e^{-\mu})x$. Therefore $V_{t+1}/V_t \leq 1 - (1-e^{-\mu})\bar{\ell}_t$. Using $\ln(1-y) \leq -y$ and summing yields

$$\ln V_{T+1} \leq \ln K - (1-e^{-\mu})\sum_{t=1}^{T}\bar{\ell}_t. \tag{18}$$

For the lower bound, $V_{T+1} \geq v_{T+1}^{(k^*)} = \exp(-\mu L^{(k^*)})$, so $\ln V_{T+1} \geq -\mu L^{(k^*)}$. Combining with (18) yields

$$(1-e^{-\mu})\sum_{t=1}^{T}\bar{\ell}_t \leq \ln K + \mu L^{(k^*)}. \tag{19}$$

We claim that $\mu/(1-e^{-\mu}) \leq 1 + \mu$ for $\mu \in (0,1]$. This is equivalent to $\mu \leq (1+\mu)(1-e^{-\mu})$. Define $f(\mu) = (1+\mu)(1-e^{-\mu}) - \mu = 1 - (1+\mu)e^{-\mu}$. Then $f(0) = 0$ and $f'(\mu) = \mu e^{-\mu} > 0$ for $\mu > 0$, so $f(\mu) > 0$ on $(0,1]$, confirming the claim.

It follows that $1/(1-e^{-\mu}) \leq (1+\mu)/\mu$. Dividing (19) by $(1-e^{-\mu})$ and subtracting $L^{(k^*)}$ gives

$$\sum_{t=1}^{T}\bar{\ell}_t - L^{(k^*)} \leq \frac{\ln K}{\mu} + \ln K + \mu L^{(k^*)}. \tag{20}$$

We now optimize over $\mu$. Set $\mu^* = \min\bigl(1, \sqrt{\ln K / L^{(k^*)}}\bigr)$, with $\mu^* = 1$ when $L^{(k^*)} = 0$.

For the first case where $L^{(k^*)} \geq \ln K$, we have $\mu^* = \sqrt{\ln K / L^{(k^*)}} \leq 1$, and Equation (20) gives $\frac{\ln K}{\mu^*} + \mu^* L^{(k^*)} = 2\sqrt{L^{(k^*)} \ln K}$. Adding the $\ln K$ term yields $2\sqrt{L^{(k^*)} \ln K} + \ln K \leq 2\sqrt{L^{(k^*)} \ln K} + 2\ln K$.

For the second case where $L^{(k^*)} < \ln K$, we have $\mu^* = 1$, and Equation (20) gives $\ln K + \ln K + L^{(k^*)} = 2\ln K + L^{(k^*)}$. It suffices to verify that $2\ln K + L^{(k^*)} \leq 2\sqrt{L^{(k^*)} \ln K} + 2\ln K$, which simplifies to $L^{(k^*)} \leq 2\sqrt{L^{(k^*)} \ln K}$, or equivalently $\sqrt{L^{(k^*)}} \leq 2\sqrt{\ln K}$, meaning $L^{(k^*)} \leq 4\ln K$. This holds since $L^{(k^*)} < \ln K < 4\ln K$.

Combining both cases, for all $L^{(k^*)} \geq 0$, there exists $\mu^*$ such that Hedge with learning rate $\mu^*$ satisfies

$$\sum_{t=1}^{T} \bar{\ell}_t - L^{(k^*)} \leq 2\sqrt{L^{(k^*)} \ln K} + 2\ln K. \tag{21}$$

The learning rate $\mu^*$ in Equation (21) depends on $L^{(k^*)}$, which requires all $T$ rounds to compute, so Hedge with $\mu^*$ is not an online algorithm. This educational derivation thus produces a small-loss bound with constants $(2, 2)$ in the $\sqrt{\cdot}$ and $\ln K$ terms via a non-online learning rate, whereas the actual statement of Lemma 3 provides the online AdaHedge bound with constants $(2, \frac{16}{3})$ plus an additive 2, taken from De Rooij et al. (2014), Theorem 8. The educational bound is qualitatively the same and is included only to make the small-loss mechanism transparent.

## B  Proofs of Theorems

### B.1  Proof of Theorem 1

*Proof.* By convexity, $f_t(x_t) - f_t(x_t^*) \leq \langle g_t, x_t - x_t^* \rangle$. Applying Lemma 1, specifically inequality (5), with $u = x_t^*$ and using $\|g_t\| \leq G$ yields

$$\eta\bigl[f_t(x_t) - f_t(x_t^*)\bigr] \leq \frac{\eta^2 G^2}{2} + \frac{1}{2}\|x_t^* - x_t\|^2 - \frac{1}{2}\|x_t^* - x_{t+1}\|^2 + \alpha\|x_t^* - \tilde{x}_t\|^2.$$

Summing over $t = 1, \ldots, T$ gives

$$\eta\mathrm{DReg}_T \leq \frac{\eta^2 G^2 T}{2} + \sum_{t=1}^{T}\Bigl(\frac{1}{2}\|x_t^* - x_t\|^2 - \frac{1}{2}\|x_t^* - x_{t+1}\|^2\Bigr) + \alpha E_T.$$

By Lemma 2, the telescoping sum is at most $\frac{D^2}{2} + DP_T = \frac{D^2 + 2DP_T}{2}$. Dividing by $\eta$ gives Equation (8). $\square$

### B.2  Proof of Theorem 2

*Proof.* The argument parallels Theorem 1 but uses the tighter inequality (4) from Lemma 1 instead of (5). Setting $u = x_t^*$ in (4) and using $\|g_t\| \leq G$, then summing over $t$ yields

$$\eta\mathrm{DReg}_T \leq \frac{\eta^2 G^2 T}{2} + \sum_{t=1}^{T}\Bigl[\frac{1}{2}\|x_t^* - x_t\|^2 - \bigl(\frac{1}{2} + \alpha\bigr)\|x_t^* - x_{t+1}\|^2\Bigr] + \alpha E_T - \alpha S_1. \tag{22}$$

We split the bracketed sum:

$$\sum_{t=1}^{T}\Bigl[\frac{1}{2}\|x_t^* - x_t\|^2 - \bigl(\frac{1}{2} + \alpha\bigr)\|x_t^* - x_{t+1}\|^2\Bigr] = \sum_{t=1}^{T}\Bigl[\frac{1}{2}\|x_t^* - x_t\|^2 - \frac{1}{2}\|x_t^* - x_{t+1}\|^2\Bigr] - \alpha S_2.$$

By Lemma 2, the first part is at most $\frac{D^2 + 2DP_T}{2}$. Substituting into (22) gives

$$\eta\mathrm{DReg}_T \leq \frac{\eta^2 G^2 T}{2} + \frac{D^2 + 2DP_T}{2} - \alpha(S_1 + S_2 - E_T) = \frac{\eta^2 G^2 T}{2} + \frac{D^2 + 2DP_T}{2} - \alpha B_T.$$

Dividing by $\eta$ yields Equation (9). Non-negativity follows from $\mathrm{DReg}_T \geq 0$, which implies $\frac{\eta T G^2}{2} + \frac{D^2 + 2DP_T}{2\eta} \geq \frac{\alpha}{\eta} B_T$. $\square$

### B.3 Proof of Theorem 3

*Proof.* Setting $\alpha = 0$ in Theorem 1 gives $\mathrm{DReg}_T \leq \phi(\eta)$, where $\phi(\eta) = \frac{\eta T G^2}{2} + \frac{D^2 + 2DP_T}{2\eta}$. The minimizer is $\eta^* = \sqrt{\frac{D^2 + 2DP_T}{TG^2}}$. Evaluating each term yields

$$\frac{\eta^* T G^2}{2} = \frac{1}{2}\sqrt{TG^2(D^2 + 2DP_T)} = \frac{R^*}{2}, \tag{23}$$

$$\frac{D^2 + 2DP_T}{2\eta^*} = \frac{1}{2}\sqrt{TG^2(D^2 + 2DP_T)} = \frac{R^*}{2}. \tag{24}$$

Hence, $\phi(\eta^*) = R^*$. $\qquad\square$

### B.4 Proof of Theorem 4

*Proof.* Since $f_t$ is $G$-Lipschitz and FTP sets $x_t = \tilde{x}_t$, we have $f_t(x_t) - f_t(x_t^*) = f_t(\tilde{x}_t) - f_t(x_t^*) \leq G\|\tilde{x}_t - x_t^*\|$. Summing over $t$ gives $\mathrm{DReg}_T \leq G\bar{E}_T$. $\qquad\square$

### B.5 Proof of Theorem 5

*Proof.* By the convexity of $f_t$ and the definition of the combined decision $x_t = \sum_k w_t^{(k)} x_t^{(k)}$, we have $f_t(x_t) \leq \sum_k w_t^{(k)} f_t(x_t^{(k)}) = m_t + GD\bar{\ell}_t$, where $m_t = \min_k f_t(x_t^{(k)})$ and $\bar{\ell}_t = \sum_k w_t^{(k)} \ell_t^{(k)}$. Define $M_T = \sum_{t=1}^T [m_t - f_t(x_t^*)] \geq 0$ (non-negative since $m_t = \min_k f_t(x_t^{(k)}) \geq f_t(x_t^*)$). Then

$$\mathrm{DReg}_T \leq GD \sum_{t=1}^T \bar{\ell}_t + M_T, \qquad \mathrm{DReg}_T^{(k^*)} = GD \sum_{t=1}^T \ell_t^{(k^*)} + M_T. \tag{25}$$

Subtracting and applying the standard Hedge bound $\sum_t \bar{\ell}_t - \sum_t \ell_t^{(k^*)} \leq \sqrt{T \ln N / 2}$ yields

$$\mathrm{DReg}_T \leq \mathrm{DReg}_T^{(k^*)} + GD\sqrt{T \ln N / 2}. \tag{26}$$

It remains to bound $\mathrm{DReg}_T^{(k^*)}$ for the best expert in the pool.

For the Type-A experts, by the geometric spacing, there exists an index $i^\dagger$ such that $\eta_{i^\dagger} \in [\eta^*/2, \eta^*]$. To see this, recall $\eta_i = 2^{-i} D/G$ for $i = 0, \ldots, K_A - 1$ where $K_A = \lceil \log_2 T \rceil + 1$, and $\eta^* = \sqrt{(D^2 + 2DP_T)/(TG^2)} \leq D/G = \eta_0$. Since $\eta_{K_A - 1} = 2^{-(K_A - 1)} D/G \leq D/(GT)$ and $\eta^* \geq D/(G\sqrt{T}) \geq \eta_{K_A - 1}$, the value $\eta^*$ lies in $[\eta_{K_A - 1}, \eta_0]$. By the dyadic spacing, there exists $i^\dagger$ with $\eta_{i^\dagger} \in [\eta^*/2, \eta^*]$. Define $\phi(\eta) = \frac{\eta T G^2}{2} + \frac{D^2 + 2DP_T}{2\eta}$, the OGD regret bound from Theorem 1 with $\alpha = 0$. Since $\phi(\eta)$ is decreasing on $(0, \eta^*]$, we have $\phi(\eta_{i^\dagger}) \leq \phi(\eta^*/2)$. Computing $\phi(\eta^*/2)$ term by term using Equations (23) and (24) yields

$$\frac{(\eta^*/2) T G^2}{2} = \frac{1}{2} \cdot \frac{R^*}{2} = \frac{R^*}{4}, \qquad \frac{D^2 + 2DP_T}{2(\eta^*/2)} = 2 \cdot \frac{R^*}{2} = R^*.$$

Therefore $\phi(\eta^*/2) = \frac{5}{4} R^*$, so the best Type-A expert satisfies $\mathrm{DReg}_T^{(A, i^\dagger)} \leq \frac{5}{4} R^*$.

For the Type-B expert, by Theorem 4, $\mathrm{DReg}_T^{(\mathrm{FTP})} \leq G\bar{E}_T$. Taking $k^*$ to be whichever expert achieves the smaller bound and substituting into Equation (26) gives Equation (11). $\qquad\square$

### B.6 Proof of Theorem 6

*Proof.* Recall the notation from the proof of Theorem 5: $M_T = \sum_{t=1}^T [m_t - f_t(x_t^*)] \geq 0$, $\bar{\ell}_t = \sum_k w_t^{(k)} \ell_t^{(k)}$, and $\phi(\eta) = \frac{\eta T G^2}{2} + \frac{D^2 + 2DP_T}{2\eta}$. From Equation (25), subtracting the expressions for $\mathrm{DReg}_T$ and $\mathrm{DReg}_T^{(k^*)}$ gives

$$\mathrm{DReg}_T - \mathrm{DReg}_T^{(k^*)} = GD\Big(\sum_{t=1}^T \bar{\ell}_t - \sum_{t=1}^T \ell_t^{(k^*)}\Big). \tag{27}$$

By Lemma 3 with $K = N$, we have

$$\sum_{t=1}^{T} \bar{\ell}_t - \sum_{t=1}^{T} \ell_t^{(k^*)} \leq 2\sqrt{L^{(k^*)} \ln N} + \tfrac{16}{3} \ln N + 2. \tag{28}$$

Since $m_t \geq f_t(x_t^*)$ (as $m_t$ is the minimum over all experts and each $f_t(x_t^{(k)}) \geq f_t(x_t^*)$), we have $L^{(k^*)} = \frac{1}{GD}\sum_{t=1}^{T}[f_t(x_t^{(k^*)}) - m_t] \leq \frac{\mathrm{DReg}_T^{(k^*)}}{GD}$. Substituting this into Equation (28) yields

$$\sum_{t=1}^{T} \bar{\ell}_t - \sum_{t=1}^{T} \ell_t^{(k^*)} \leq 2\sqrt{\frac{\mathrm{DReg}_T^{(k^*)} \ln N}{GD}} + \tfrac{16}{3} \ln N + 2. \tag{29}$$

Combining Equations (27) and (29) gives

$$\mathrm{DReg}_T \leq \mathrm{DReg}_T^{(k^*)} + 2\sqrt{GD \cdot \mathrm{DReg}_T^{(k^*)} \cdot \ln N} + \tfrac{16}{3} GD \ln N + 2GD. \tag{30}$$

Define $\psi(x) = x + 2\sqrt{GDx \ln N} + \frac{16}{3} GD \ln N + 2GD$. For $x > 0$, $\psi'(x) = 1 + \sqrt{GD \ln N/x} > 0$, so $\psi$ is increasing on $[0, \infty)$. By the same argument as in Theorem 5, the best expert satisfies $0 \leq \mathrm{DReg}_T^{(k^*)} \leq D^*$. Since $\psi$ is increasing, $\psi(\mathrm{DReg}_T^{(k^*)}) \leq \psi(D^*)$, which yields Equation (12). $\qquad\square$

### B.7 Proof of Theorem 7

*Proof.* Recall $\phi(\eta) = \frac{\eta T G^2}{2} + \frac{D^2 + 2DP_T}{2\eta}$, $M_T = \sum_{t=1}^{T}[m_t - f_t(x_t^*)]$, and $D^{**} = \min\{\frac{5}{4}R^*, G\bar{E}_T, \frac{5}{4}R^* - \frac{\alpha}{\eta_{j\dagger}} B_T^{(j\dagger)}\}$ as defined in Equation (14). By Theorem 2, the Type-C expert with parameters $(\eta_{j\dagger}, \alpha)$ satisfies

$$\mathrm{DReg}_T^{(C,j\dagger)} \leq \phi(\eta_{j\dagger}) - \frac{\alpha}{\eta_{j\dagger}} B_T^{(j\dagger)} \leq \frac{5}{4} R^* - \frac{\alpha}{\eta_{j\dagger}} B_T^{(j\dagger)}, \tag{31}$$

where the second inequality follows from $\phi(\eta_{j\dagger}) \leq \phi(\eta^*/2) = \frac{5}{4}R^*$, as established in the proof of Theorem 5. We verify that $D^{**} \geq 0$. The first two terms in Equation (14) are non-negative. For the third, since $\mathrm{DReg}_T^{(C,j\dagger)} \geq 0$ (as dynamic regret is always non-negative), Equation (31) implies $\frac{5}{4}R^* - \frac{\alpha}{\eta_{j\dagger}} B_T^{(j\dagger)} \geq \mathrm{DReg}_T^{(C,j\dagger)} \geq 0$.

The expert $j^\dagger$ is one specific expert in the pool. The dynamic regret of the best expert among all $N'$ is at most $D^{**}$. Repeating the proof of Theorem 6 with $N'$ in place of $N$ and $D^{**}$ in place of $D^*$, the monotonicity of $\psi$ yields Equation (13). $\qquad\square$

