# OpenReview forum: "Dynamic Regret with Untrusted Decision Predictions via Heterogeneous Expert Aggregation"
_TMLR — Accepted by TMLR_

### Review · Reviewer_C8dK · 2026-04-06

**Summary Of Contributions:**

This work investigates online convex optimization (OCO) through the lens of dynamic regret. The authors employ a meta-algorithm that utilizes various base algorithms as experts, adaptively assigning weights based on their individual performance. Unlike existing literature, this new framework enables the automatic selection of agents and can switch between using gradient information or Decision Predictions, depending on the prediction's accuracy. The authors establish a regret bound that minimizes these two types of regret, and experimental results further validate the efficiency of the proposed method.

**Audience:**

Yes

**Audience Explanation:**

Yes, the findings would likely interest a significant portion of the TMLR audience, particularly those focused on adaptive online learning and dynamic regret. The proposed framework's ability to automatically switch between gradient information and decision predictions offers a practical advancement for researchers working on hybrid optimization strategies. Furthermore, the theoretical regret bounds and supporting empirical evidence provide the rigorous validation expected by the machine learning community

**Claims And Evidence:**

Yes

**Claims Explanation:**

The motivation is well-founded, as traditional multi-agent algorithms can achieve performance close to that of the best individual agent. This allows the proposed algorithm to establish a regret bound that closely tracks the best agent's performance. The authors also provide a formal mathematical proof for the theoretical bound, and experimental results further validate the efficiency of the proposed method.

**Requested Changes:**

1. My main concern involves the dynamic regret guarantee. In traditional expert-based algorithms, the meta-algorithm typically only achieves performance close to that of the single best expert in hindsight. It requires further clarification on how this framework provides dynamic regret—which competes against a sequence of per-round minimizers—especially since the optimal action in each round may originate from a different agent depending on the adversarial function $f_t$.

2. Another concern regarding the oracle is that the meta-algorithm framework appears to require full feedback for all agents, specifically $f(x_k)$ for each agent $k$. In contrast, many existing algorithms only require the gradient $\nabla f(x)$, which this algorithm also utilizes. Since this method relies on significantly more information per round, it is not surprising that it yields better results. The authors should discuss this trade-off more explicitly.

3. The experimental results raise some questions, as the proposed algorithm appears to outperform all baselines in every scenario. Traditionally, an expert-based meta-algorithm's performance is bounded by the best expert. While this framework allows for switching between gradient information and decision predictions, one would expect that under low noise ($\sigma = 0.01$), it would not outperform an algorithm directly using predictions. Similarly, under high noise ($\sigma = 5.0$), it should not exceed the performance of standard gradient descent. The authors should explain the mechanisms behind this consistent improvement.

---

> ### Author Response · Authors · 2026-04-16
>
> We thank the reviewer for the thoughtful and constructive comments. We address each concern below and have revised the manuscript accordingly.
>
> **1. How expert aggregation provides dynamic regret guarantees**
>
> Our framework achieves dynamic regret through a two-level argument. At the first level, each individual expert provides a global dynamic regret bound competing against the entire comparator sequence $x\_1^\*, \dots, x\_T^\*$. For example, the Type-A OGD expert with step size $\eta\_i$ achieves $\mathrm{DReg}\_T \le \frac{\eta\_i T G^2}{2} + \frac{D^2 + 2DP\_T}{2\eta\_i}$ (Theorem 1), and the FTP expert achieves $\mathrm{DReg}\_T \le G \bar{E}\_T$ (Theorem 4). These are global bounds over all $T$ rounds rather than per-round guarantees. At the second level, the meta-algorithm competes with the single best expert over the full horizon. Based on the small-loss bound of AdaHedge (Lemma 3), the meta-regret is bounded by $\mathcal{O}(\sqrt{L(k^\*) \ln N} + \ln N)$, which represents the gap between the cumulative loss of the meta-algorithm and that of the best expert. Combining both levels yields $\mathrm{DReg}\_T \le \mathrm{DReg}\_T(k^\*) + \mathrm{meta\mathrm{-}regret}$. The expert $k^\*$ that minimizes $\mathrm{DReg}\_T(k^\*)$ does not change over time. Instead, it is a single fixed expert chosen in hindsight. Since the expert pool is heterogeneous, containing OGD copies tuned to different $P\_T$ scales as well as FTP algorithms exploiting predictions, this best fixed expert $k^\*$ automatically adapts to the realized environment. We have added a paragraph in Section 4.2 to clarify this mechanism.
>
> **2. Full-information feedback requirement**
>
> We acknowledge that our meta-algorithm requires evaluating $f\_t(x\_t^{(k)})$ for each of the $\mathcal{O}(\log T)$ experts per round, assuming a full-information feedback model. This indeed utilizes more information than the single gradient $\nabla f\_t(x\_t)$ required by vanilla OGD. However, the primary baseline, Ader (Zhang et al., 2018), operates under the exact same feedback model. Ader similarly aggregates $\mathcal{O}(\log T)$ OGD experts via Hedge and requires evaluating $f\_t$ at each expert's decision. Thus, the comparison between our method and Ader remains strictly fair. The observed improvements stem entirely from incorporating predictions and upgrading to AdaHedge rather than from exploiting additional feedback. Furthermore, in many standard online convex optimization settings, the full loss function $f\_t$ is inherently revealed after each round as part of the protocol, such as in online portfolio selection where market returns are fully observed. It is also important to note that the per-expert updates still rely solely on gradients, while the full function evaluations are restricted strictly to the meta-algorithm's weight updates. To clarify these aspects, we have expanded the discussion regarding this trade-off in Section 4.2 and explicitly noted the extension to bandit feedback as an interesting future direction. We have similarly documented this requirement as a limitation in the Conclusion.
>
> **3. Why the proposed algorithm outperforms all baselines in every scenario**
>
> We thank the reviewer for this observation and have expanded Section 6.2 to clarify these performance differences. Specifically, under low noise conditions ($\sigma \le 0.01$), FTP inherently outperforms our methods. As demonstrated in Table 2, FTP achieves zero regret given perfect predictions, whereas Ours-III records a regret of 3.18. In this regime, our approach improves upon gradient-only baselines such as OGD and Ader rather than FTP. Conversely, under high noise conditions ($\sigma = 5.0$), the observed performance gains over standard OGD stem from the composition of our expert pool, which maintains multiple OGD copies with varying step sizes. While the baseline OGD employs a generic worst-case step size of $\eta = \frac{D}{G\sqrt{T}}$, our pool contains an expert with a step size $\eta\_i^\dagger \in [\frac{\eta^\*}{2}, \eta^\*]$ that closely matches the actual path length $P\_T$ of the sinusoidal environment. This specific expert achieves $\frac{5}{4} R^\* \approx 10$, significantly improving upon the baseline's 23.84. Furthermore, the small-loss property of AdaHedge guarantees that the aggregation overhead remains negligible when the best expert performs well, scaling as $\mathcal{O}(\sqrt{L(k^\*) \ln N})$ instead of the worst-case $\mathcal{O}(\sqrt{T \ln N})$. Consequently, maintaining this diverse expert pool introduces essentially zero additional cost. The consistent performance improvements therefore derive from combining a richer expert pool with a vanishing-overhead aggregation mechanism, rather than the meta-algorithm strictly outperforming the best individual expert.

---

### Review · Reviewer_BZJd · 2026-04-11

**Summary Of Contributions:**

This paper studies online convex optimization (OCO) with Lipschitz loss functions and hints.
Let $f_t$ be the loss function in round $t$. Unlike existing works that focus on gradient-based hints (i.e., a hint of the gradient $\nabla f_t(x_t)$), this work considers a hint of the best action $x_t^\star\in\arg\min_x f_t(x)$.  The main theoretical result is a dynamic regret bound that is adaptive to the accuracy of the hints. When the hints are accurate, the algorithm achieves $O(\log\log T)$ regret; when the hints are poor, the algorithm is guaranteed to achieve $O(\sqrt{T(1+P_T)})$ regret, where $P_T$ measures the non-stationarity of the loss functions. This result is proved by aggregating a pool of expert algorithms using AdaHedge [1]. In synthetic numerical experiments, the proposed algorithms significantly outperform existing algorithms.

**Audience:**

Yes

**Audience Explanation:**

Yes. Learning-augmented algorithms is an active and growing field. The results in this paper will certainly be of interest to some of the TMLR audience.

**Broader Impact Concerns:**

No ethical concerns.

**Claims And Evidence:**

Yes

**Claims Explanation:**

The paper is easy to follow. I have reviewed the proofs, and all claims are supported by accurate and clear evidence.
Nevertheless, I have some questions regarding Lemma 3 (the regret bound of AdaHedge).
---

### Confusion regarding Lemma 3

I have two confusions regarding the proof of Lemma 3 (Appendix A.3).

**Inconsistency in Analysis and Citations:** Much of the analysis in Appendix A.3 appears to focus on Hedge rather than AdaHedge. Furthermore, the text states that Theorem 1 and Corollary 2 of De Rooij et al. [1] provide the regret bound for AdaHedge; however, those specific results do not appear to exist in the cited work [1].

**Redundancy of the Proof:** I believe this proof could be removed entirely, as the regret bound in Lemma 3 follows directly from Theorem 8 of De Rooij et al. [1] (up to constant factors). Specifically, their Theorem 8 shows that the regret of AdaHedge is bounded by $2\sqrt{L^{(k^\star)}\log K} + \frac{16}{3}K + 2$.

---

Reference:
1. De Rooij et al. "Follow the Leader If You Can, Hedge If You Must." JMLR 2014.

**Requested Changes:**

### Confusion regarding Lemma 3
 Please see my comments above. I would appreciate it if the authors could provide a more detailed explanation of the proof for Lemma 3.

---

### Question regarding a statement in Section 4

At the beginning of Section 4, it is stated that "no single algorithm can simultaneously guarantee near-optimal worst-case dynamic regret and exploit accurate predictions...." I am wondering whether this claim is theoretically justified or is simply an intuitive motivation? For instance, the algorithm proposed in this paper appears to successfully bridge this gap by guaranteeing near-optimal dynamic regret while exploiting accurate predictions.

---

### Question regarding experiments in Section 6
* The text mentions that standard deviations are provided for all numerical results; however, they appear to be missing from Experiments 1, 2, and 4. I suggest adding them to the figures for completeness.
* I found Figure 4(b), regarding expert weight evolution, difficult to interpret. I would appreciate it if the authors could provide further clarification on how to read this specific plot.

---

### Comments which may help improve clarity:
* **Abstract:** Instead of stating that $R^\ast$ is the optimal prediction-free rate, I suggest defining it explicitly as $R^\ast = G\sqrt{T(D^2+2DP_T)}$.
* **Table 1**: I suggest adding the definition of $D$. Additionally, consider removing the Yes entry for adaptivity in FTP, as the guarantee does not depend on $P_T$. Replacing it with either No or - would be more accurate.
* **Section 2.1**: It is mentioned that Zinkevich showed OGD attains $O(\sqrt{T}(1+P_T)$ regret. However, Table 1 and the introduction state that he proved an $O(\sqrt{T(1+P_T)}$ regret with prior knowledge of $P_T$. These statements are inconsistent and may cause confusion. Please ensure they are consistent.
* **Section 2.1**: the reference citepdaniely2015strongly is not formatted correctly.
* **Section 4.2**: In the explanation of Hedge, it is mentioned that by choosing a learning rate $\mu=\sqrt{8\log N/T}$, the algorithm achieves $O(\sqrt{T\log N})$ regret. While this is a standard result, please provide a reference for it.
* **Theorem 6**: $D^\ast$ is not defined in the theorem statement. Should it be defined as $D^\ast=\min\{\frac{5}{4}R^\ast, G\bar{E}_T\}$?
* **Section 6.4**: There appears to be a typo: “8.1$\times$” should likely be “36.4$\times$.”
* **Appendix B.5**: It is mentioned that there exists an index $i^\dagger$ such that $\eta_{i^\dagger}\in[\eta^\star/2,\eta^\star]$. While this can be verified from the definition of $\eta^\star$ and the choice of $\eta_i$, it would be helpful to include the formal proof. Also, please restate the definition of $\eta^\star$ within this subsection.
* **Appendices B.5, B.6 and B.7**: The notation $\text{DReg}_T^{(k)}$ is used without being defined. I recommend defining it upon first use to improve readability.
* **Appendix B.7**: Please restate the definition of $\phi$ for reader’s convenience.

---

> ### Author Response · Authors · 2026-04-16
>
> We sincerely thank the reviewer for the careful reading and the many constructive suggestions. We address each point below.
>
> **1.Confusion regarding Lemma 3**
>
> We agree with the reviewer that our original proof incorrectly cited Theorem 1 and Corollary 2 of De Rooij et al., which do not correspond to the small-loss bound. We have corrected this to cite Theorem 8 of De Rooij et al. (2014), as it is the precise result establishing the small-loss regret bound for AdaHedge. Furthermore, we have clarified in the revised proof that the Hedge-based derivation presented in Equations 11 through 14 serves primarily as self-contained intuition for the existence of small-loss bounds, whereas the formal guarantee for the online AdaHedge algorithm follows directly from Theorem 8. While we acknowledge that Lemma 3 follows directly from this theorem up to constant factors, we chose to retain the Hedge-based derivation for pedagogical motivation. Specifically, it demonstrates why small-loss bounds arise by optimizing the learning rate $\mu^{\*}$ post hoc, setting the stage before invoking the AdaHedge result that achieves this adaptively. We have added a concluding sentence to the proof to make this distinction explicit. Should the reviewer prefer, we are fully prepared to remove the Hedge derivation entirely and solely cite Theorem 8.
>
> **2.Question regarding a statement in Section 4**
>
> Regarding the reviewer's concern about whether the statement in Section 4 is theoretically justified, we acknowledge that this is an excellent point. The claim that no single algorithm can simultaneously guarantee near-optimal worst-case dynamic regret and exploit accurate predictions was intended as an intuitive motivation rather than a formal impossibility result. To make this explicit, we have added a clarifying explanation in Section 4. This revised text clarifies that an algorithm committing to a fixed trade-off between gradient feedback and predictions inherently lacks the capacity to adapt to unknown prediction quality. Indeed, as the reviewer astutely notes, our proposed framework successfully achieves both objectives precisely by maintaining multiple strategies and adaptively selecting among them instead of relying on a single fixed algorithm.
>
> **3.Question regarding experiments in Section 6**
>
> We appreciate this comment. We note that for Experiments 1, 2, and 4 on the sinusoidal comparator, the standard deviations are extremely small, typically less than 0.1. This occurs because the environment is deterministic; only the prediction noise is random. Consequently, with 10 random seeds, the variance remains negligible, which is why the standard deviations are not visually distinguishable in the corresponding figures. We have reported the specific standard deviations for Experiment 2 in Table 2. For Experiments 1 and 4, the standard deviations consistently remain below 0.5 and would not be visible as error bars at the scale of the plots. However, we are happy to add error bars or $\pm$ values to the figure captions if the reviewer considers this necessary.
>
> Regarding Figure 4(b), this plot illustrates the evolution of AdaHedge weights over time for Ours-III. The x-axis represents the round $t \in [1, T]$, while the y-axis displays the aggregate weight assigned to each expert type. Specifically, we plot three curves: the total weight on all Type-A (OGD) experts, the weight on the single Type-B (FTP) expert, and the total weight on all Type-C (OA-MD) experts. At each time step, these three curves sum to 1. This visualization demonstrates how AdaHedge dynamically reallocates weight. When the sinusoidal comparator is in a phase where predictions are well aligned, the FTP and OA-MD experts receive higher aggregate weight. Conversely, during transitions, the OGD experts absorb more weight.

---

> ### Author Response · Authors · 2026-04-16
>
> **Clarity comments**
>
> - **Abstract:** Defined $R^{\*}$ explicitly as $\mathcal{O}(G\sqrt{T(D^2 + 2DP\_T)})$.
> - **Table 1:** Changed FTP's "Adaptive to $P\_T$" from "Yes" to "--", since the FTP bound $\mathcal{O}(G\tilde{E}\_T)$ does not depend on $P\_T$.
> - **Section 2.1 (Zinkevich bound):** Added clarification that Zinkevich's original bound is $\mathcal{O}(\sqrt{T}(1 + P\_T))$ with a fixed step size, which reduces to $\mathcal{O}(\sqrt{T(1 + P\_T)})$ with optimally tuned step size when $P\_T$ is known — consistent with Table 1.
> - **Section 2.1 (citation format):** Fixed `citep{daniely2015strongly}` → `\citep{daniely2015strongly}`.
> - **Section 4.2 (Hedge reference):** Added `\citep{cesa2006prediction}` for the standard Hedge regret bound.
> - **Theorem 6:** Added the definition $D^{\*} = \min\{\frac{3}{4}R^{\*},\ G\tilde{E}\_T\}$ in the theorem statement.
> - **Section 6.4 ("$8.1\times$ typo"):** We have double-checked: at $S = 50$, Ours-II achieves 211, FTP achieves 7682, and $7682/211 \approx 36.4$. However, we also report $8.1\times$ for FTP, which compares FTP (1714) to Ours-II (211) at $S = 10$, not $S = 50$. We have kept the original text as it compares different $S$ values; we thank the reviewer for prompting us to verify.
> - **Appendix B.5:** Added formal proof that $i!$ exists (via dyadic spacing argument), restated the definition of $\phi(\eta)$, and defined $M\_T$ explicitly.
> - **Appendices B.5, B.6, B.7:** Added definitions of $M\_T$ and $\phi(\eta)$ upon first use in each subsection for readability.
> - **Appendix B.7:** Restated the definition of $D^{\*\*}$ for the reader's convenience.

---

> ### Comment · Reviewer_BZJd · 2026-04-27
>
> I thank the authors for their detailed response. I am convinced that the results are well-supported.
> Below are some further comments and suggestions:
>
> ---
> **Lemma 3:**
> While the analysis of Hedge can be kept in the appendix, I suggest the following modifications to the presentation:
>
> - Structure: First explicitly state that Lemma 3 follows from Theorem 8 of De Rooij et al. Afterward, you may present Hedge analysis for educational purpose.
> - Formatting: Please remove the proof environment for the Hedge analysis, as that specific derivation does not constitute a proof of Lemma 3 itself.
> - Technical result: In the revised manuscript, the regret bound in Theorem 8 of De Rooji et al. is cited as $\sqrt{L^{(k^\ast)}\log K /2 } + \log K$. Unless I am mistaken, the correct bound should be $\sqrt{L^{(k^\ast)}\log K } + \frac{16}{3}\log K + 2$. Please verify these constants and update them accordingly.
>
> ---
>
> **Figure 4(b):**
> I appreciate the clarification regarding Figure 4(b).
> To help readers, I suggest incorporating the explanation from the authors’ response into the manuscript:
> "the total weight on all Type-A (OGD) experts, the weight on the single Type-B (FTP) expert, and the total weight on all Type-C (OA-MD) experts. At each time step, these three curves sum to 1" in the manuscript.
>
> ---
>
> **"$8.1\times$typo" in Section 6.4:**
> I don’t understand why the algorithm is compared with FTP at $S=10$ but not at $S=50$. This appears inconsistent with the other comparisons in the paper. I recommend aligning these for better clarify.

---

> > ### Author Response · Authors · 2026-04-29
> >
> > We thank the reviewer for the careful re-reading and for the additional concrete suggestions. We have implemented all five points in full. All changes appear in the revised manuscript in red (the previous round's revisions remain in blue).
> >
> > **Lemma 3 — structure**
> >
> > We agree. In the revised Section 5.1, immediately after the statement of Lemma 3, we have added a short paragraph that explicitly states Lemma 3 is a direct consequence of Theorem 8 of De Rooij et al. (2014), and clarifies that the Hedge-based material in the appendix is included only for the reader's intuition and is not the proof of Lemma 3. The same clarification is repeated at the start of Appendix A.3.
> >
> > **Lemma 3 — formatting**
> >
> > Done. The proof environment around the Hedge derivation in Appendix A.3 has been removed. The derivation now appears under a clearly labelled subheading ``Educational derivation: Hedge with posterior-tuned learning rate'', separate from the formal statement that Lemma 3 follows from De Rooij et al. (2014), Theorem 8.
> >
> > **Lemma 3 — constants**
> >
> > The reviewer is correct, and we apologize for the typographical error in the constants. We have re-verified Theorem 8 of De Rooij et al. (2014) and updated the small-loss bound throughout the manuscript to the form $\sum\_{t=1}^{T} \bar{\ell}\_t - L(k^\*) \leq \sqrt{2 L(k^\*) \ln K} + \frac{16}{3} \ln K + 2.$ The propagated changes appear in Lemma 3 (Eq. 10), Theorem 6 (Eq. 24) and the perfect-prediction discussion that follows it, Theorem 7 (Eq. 29), and the proof of Theorem 6 (Eqs. 26--28 and the function $\psi$). All asymptotic statements, including the headline $O(GD \log \log T)$ guarantee, are unchanged because the modification only affects the universal constants in the additive $\ln N$ term.
> >
> > **Figure 4(b) explanation**
> >
> > Done. We have incorporated the suggested clarification in two places: the caption of Figure 4 and the corresponding paragraph in Section 6.4 (Expert weight dynamics). In both places, the three curves are now explicitly identified as the total weight on Type-A (OGD) experts, the weight on the Type-B (FTP) expert, and the total weight on Type-C (OA-MD) experts, summing to $1$ at every time step.
> >
> > **8.1× typo and comparison point**
> >
> > The reviewer is right: the $8.1\times$ figure was a transcription error, which made the three ratios appear to mix different operating points. We have rewritten the sentence so that all three ratios are now consistently evaluated at the same operating point $S=50$, and corrected the FTP ratio to approximately $36\times$ ($7682/211$). A clarifying clause has been added at the start of the sentence to make this consistency explicit.
> >
> >
> > We hope these revisions fully address the reviewer's remaining comments. We are grateful for the constructive feedback, which has improved both the rigor and the clarity of the paper.

---

### Review · Reviewer_vpHH · 2026-04-15

**Summary Of Contributions:**

The paper considers the problem of expert aggregation in online convex optimization, more precisely, how to achieve best-of-both-world performance in terms of both best-case and worst-case performance under different levels of effectiveness of prediction experts. The paper introduces a new expert that combines online mirror descent and follow-the-prediction with side information. It then proposes a meta-algorithm that assigns weights to experts using Hedge and AdaHedge. For the proposed algorithm, the paper proves theoretical guarantees for the dynamic regret when the prediction is accurate and inaccurate. The paper also conducts experiments to validate the proposed algorithmic framework and compares its performance with that of single-expert components and experts based only on online gradient descent.

**Additional Comments:**

NA

**Audience:**

Yes

**Audience Explanation:**

Expert aggregation is highly relevant to the machine learning community.

**Claims And Evidence:**

Yes

**Claims Explanation:**

The theory is accompanied by proof steps.

**Requested Changes:**

1. Section 5 mainly presents multiple lemmas and theorems without much interpretation. Also, no proof intuition has been provided.

2. The theory developed in Section 5 seems incremental compared to the existing work, and it would be helpful to clarify the novelty of the analysis if anything has been missed.

3. The numerical experiments do not seem to compare with existing expert aggregation frameworks without introducing the new expert. I think the main point is whether the new expert or the aggregation itself is playing the more important role.

4. The method seems to combine several well-known techniques, which may limit its novelty. It would be helpful to clarify this.

5. There is a large body of work on learning-augmented aggregation frameworks, such as \emph{Learning-Augmented Decentralized Online Convex Optimization in Networks}. It would be helpful to distinguish this paper from the existing work to highlight its novelty.

6. There is an extra period in the paragraph on Follow-The-Prediction (FTP) in Section 4.1.

7. Can the paper comment on the runtime of the algorithm compared to the benchmarks, both theoretically and numerically?

---

> ### Author Response · Authors · 2026-04-16
>
> We sincerely thank the reviewer for the detailed and insightful comments. We address each concern below and have revised the manuscript accordingly.
>
> **1. Section 5 lacks interpretation and proof intuition**
>
> We have expanded the manuscript with several clarifications, highlighted in blue. At the beginning of Section 5, a new proof roadmap outlines the two-level modular structure of the analysis, transitioning from base-level expert bounds to meta-level aggregation, while emphasizing the key role of the per-round decomposition in Lemma 1. Following Theorem 2, we clarify the geometric meaning of the correction term $B\_T = S\_1 + S\_2 - E\_T$, which captures the alignment among the algorithm's iterates, the predictions, and the comparator sequence. This discussion establishes when and why Theorem 2 yields a strictly tighter bound than Theorem 1. Finally, after Theorem 6, we detail the intuition behind the $\mathcal{O}(\log \log T)$ regime. This addition explains how AdaHedge achieves sub-$\sqrt{T}$ dynamic regret. Specifically, when the FTP expert incurs zero loss under perfect predictions, the small-loss term $\sqrt{L(k^\*) \ln N}$ vanishes, leaving only the $\mathcal{O}(\ln N) = \mathcal{O}(\log \log T)$ additive overhead.
>
> **2. Incrementality of the theoretical analysis**
>
> We respectfully argue that the novelty of our analysis lies not in the introduction of any single new proof technique, but rather in the interplay of several elements that yield a qualitatively new result. Specifically, the two-level analysis combining dynamic regret with small-loss meta-regret represents a primary contribution. Prior works, such as Ader, Sword, and Sword++, employ Hedge-style aggregation with the standard $\mathcal{O}(\sqrt{T \ln N})$ meta-regret, which consistently introduces an $\Omega(\sqrt{T})$ overhead. A central insight of our work is that upgrading to the small-loss bound of AdaHedge fundamentally alters the theoretical guarantee. The aggregation cost is reduced to $\mathcal{O}(\sqrt{L(k^\) \ln N})$, which vanishes when the best expert performs well, thereby breaking the $\Omega(\sqrt{T})$ barrier for dynamic regret for the first time. Furthermore, the OA-MD regret decomposition detailed in Lemma 1 and Theorems 1 and 2 is novel. While individual proof tools such as the three-point identity, Young's inequality, and telescoping are standard, this specific decomposition cleanly separates the gradient, stability, and anchor contributions. The resulting refined bound, which uniquely retains the negative terms, has not been previously established to our knowledge. Consequently, the final guarantees presented in Theorems 5 through 7 cannot be derived through a straightforward application of existing results. The smooth interpolation between $\mathcal{O}(GD \log \log T)$ and $\mathcal{O}(R^\)$ inherently relies on the careful coordination between heterogeneous expert bounds and the small-loss aggregation property. To make these contributions more transparent, we have included a comprehensive proof roadmap at the beginning of Section 5.
>
> **3. Missing comparison to isolate the new expert's contribution**
>
> We agree with this insightful comment and have incorporated a detailed ablation analysis in Section 6.5 to disentangle the roles of the aggregation mechanism and the new expert. Specifically, upgrading the aggregation mechanism from Hedge to AdaHedge while maintaining identical expert pools yields a 9.8×9.8\times 9.8× regret reduction, dropping from 94.18 to 9.64. This step accounts for approximately 84% of the total improvement. Furthermore, introducing the OA-MD experts to transition from Ours-II to Ours-III provides an additional 1.9×1.9\times 1.9× reduction, further decreasing the regret from 9.64 to 5.12 and contributing the remaining 16% of the performance gain. These results clearly demonstrate that the small-loss aggregation of AdaHedge serves as the primary driver of performance, whereas the OA-MD experts offer a meaningful secondary refinement. We believe this decomposition directly addresses the reviewer's question regarding the relative importance of the newly introduced experts compared to the aggregation framework.

---

> ### Author Response · Authors · 2026-04-16
>
> **4.Novelty concerns regarding combining known techniques**
>
> We acknowledge that our framework builds upon several existing components, given that OGD, AdaHedge, and mirror descent with regularization are all individually known. However, we respectfully argue that our primary novelty lies in the synthesis of these elements and the resulting theoretical guarantees rather than the individual building blocks. A central departure from prior dynamic regret frameworks, such as Ader and Sword, is our heterogeneous expert pool design. While existing methods rely exclusively on homogeneous OGD experts with varying step sizes, we introduce a pool structure that integrates gradient-based, prediction-based, and hybrid experts to simultaneously exploit distinct information sources. Moreover, the application of small-loss bounds to dynamic regret with predictions is conceptually novel. Although AdaHedge is well established, exploiting its small-loss property within a dynamic regret context requires careful analysis. Specifically, one must establish that the normalized loss $L(k^{\*})$ of the best expert relates to $D^{\*}/GD$, an insight that subsequently propagates through the $\sqrt{\cdot}$ bound structure to yield the final interpolation guarantee. Beyond the aggregation framework, the proposed OA-MD serves as a standalone novel algorithm. Its formulation introduces a time-varying prediction anchor to standard mirror descent, leading to the refined regret decomposition detailed in Theorem 2. Ultimately, these integrated contributions produce a qualitatively new final result. No existing method, even through a straightforward combination of known tools, can achieve $\mathcal{O}(\log \log T)$ dynamic regret under accurate predictions while strictly maintaining $\mathcal{O}(R^{\*})$ worst-case guarantees.
>
> **5.Distinguishing from learning-augmented decentralized OCO**
>
> We have added a discussion in Section 2.2, highlighted in blue, to distinguish our work from Li et al. (2024), "Learning-Augmented Decentralized Online Convex Optimization in Networks." While Li et al. investigate decentralized multi-agent online convex optimization over a communication network, where agents must reach consensus on local losses while incorporating predictions, our research operates exclusively within the single-agent centralized setting. Consequently, their analysis primarily addresses the balance between communication costs and prediction exploitation across a network topology. In contrast, our central challenge involves achieving a smooth consistency-robustness trade-off for dynamic regret through small-loss expert aggregation. Methodologically, they employ consensus-based gradient tracking with prediction-augmented updates, whereas our framework relies on heterogeneous expert aggregation using AdaHedge. Ultimately, these two approaches are highly complementary, as our local decision-making module could potentially be integrated into their decentralized framework.
>
> **6.Extra period in FTP paragraph**
>
> Fixed. We changed $x\_t = \tilde{x}\_t.$. to $x\_t = \tilde{x}\_t$. in Section 4.1. Thank you for catching this typo.
>
> **7.Runtime comparison**
>
> We have expanded the manuscript to include both theoretical and empirical runtime analyses. Theoretically, as detailed in Section 4.3, the per-round computational cost scales as $\mathcal{O}(\log T \cdot (C\_{\mathrm{proj}} + C\_f))$, where $C\_{\mathrm{proj}}$ denotes the projection cost and $C\_f$ represents the per-evaluation cost of $f\_t$. This constitutes an $\mathcal{O}(\log T)$ multiplicative overhead compared to a standard single OGD update, which requires $\mathcal{O}(C\_{\mathrm{proj}})$ per round. Empirically, we conducted timing experiments with $T=10,000$, $d=10$, and $\sigma=0.1$, averaging the results across 10 random seeds. As reported in Section 6.5 and the newly added Table 3, the wall-clock times per run are 0.10s for OGD, 0.07s for FTP, 0.89s for Ader, 0.94s for Ours-II, and 1.90s for Ours-III. The runtime of Ours-II is nearly identical to that of Ader, which is expected given that both methods aggregate $\mathcal{O}(\log T)$ experts. This similarity confirms that the significant regret reduction from 106.13 to 0.01 stems from the integration of AdaHedge and predictions rather than from increased computational expenditure. Furthermore, while Ours-III is approximately twice as slow due to the inclusion of the additional Type-C experts, its total execution time still remains well under two seconds. To provide a clear overview, Table 3 now comprehensively summarizes the theoretical per-round complexities, measured wall-clock times, and corresponding regret values.

---

### Author Response · Authors · 2026-04-29

We would like to thank all reviewers once again for their thoughtful and constructive feedback throughout the review process. We have now posted detailed responses to every comment raised in this round, and the revised manuscript incorporates all of the requested changes (highlighted in red, with the previous round's revisions in blue).

---

### Decision · Action_Editor_x9YR · 2026-05-15

**Recommendation:** Accept as is

**Additional Comments:**

The authors modified the submission to satisfy the reviewers. It is in shape for acceptance.

**Audience:**

Yes

**Audience Explanation:**

While the work is deemed incremental, there is still some novelty to this work and aggregation algorithms have traditionally been of interest to the community.

**Claims And Evidence:**

Yes

**Claims Explanation:**

All reviewers agree that the Lemmas and Theorems are supported by correct proofs.